# Local mapping of the nanoscale viscoelastic properties of fluid membranes by AFM nanorheology

William Trewby [1,2] ✉, Mahdi Tavakol[1,3] & Kislon Voïtchovsky [1] ✉

Biological membranes are intrinsically dynamic entities that continually adapt their biophysical properties and molecular organisation to support cellular function. Current microscopy techniques can derive high-resolution structural information of labelled molecules but quantifying the associated viscoelastic behaviour with nanometre precision remains challenging. Here, we develop an approach based on atomic force microscopy in conjunction with fast nano-actuators to map the viscoelastic response of unlabelled supported membranes with nanometre spatial resolution. On fluid membranes, we show that the method can quantify local variations in the molecular mobility of the lipids and derive a diffusion coefficient. We confirm our experimental approach with molecular dynamics simulations, also highlighting the role played by the water at the interface with the membrane on the measurement. Probing ternary model bilayers reveals spatial correlations in the local diffusion over distances of ≈20 nm within liquid disordered domains. This lateral correlation is enhanced in native bovine lens membranes, where the inclusion of protein-rich domains induces four-fold variations in the diffusion coefficient across < 100 nm of the fluid regions, consistent with biological function. Our findings suggest that diffusion is highly localised in fluid biomembranes.

Lipid membranes play a vital role in cellular function; they provide a barrier to the external environment, compartmentalise organelles and host the proteins necessary for transport, signalling and sensing[1]. The lipids themselves, far from forming a passive membrane matrix, are active participants in cellular machinery. They regulate membrane protein function[2] and folding[3], are central to the transport and storage of fats and cholesterol around the body[4], and provide conduction pathways for ions and protons[5–7]. These processes hinge on the membrane's ability to locally reorganise its molecular composition and biophysical properties depending on the task at hand. The local forces and dynamics associated with this are non-trivial due to the membrane's molecularly crowded environment[8–11]. Experimentally, in-situ tracking of mobility at the scale of the lipids themselves is challenging, being sensitive to the smallest variations in temperature, ionic content,

external fluid flow and substrate interactions[12–17]. With lipids playing a direct role in lateral membrane organisation of both mammalian[9,18–20] and bacterial[21–25] membranes, any valid picture must accommodate both the in-plane forces and resultant dynamics, as well as lateral heterogeneity on the scale of a few molecules.

Most studies to date rely on molecular mobilities obtained through measurements that average over relatively large membrane areas comprising over $10^6$ molecules. Techniques such as fluorescence recovery after photobleaching (FRAP) and fluorescence correlation spectroscopy (FCS)[26–28] track the average motion of fluorescent dyes within areas of tens to hundreds of square microns, hence providing a rather indirect, macroscopic picture of molecular mobility. This is quantified with an average diffusion coefficient, $D$, that measures the molecules' mean-squared displacement per unit time[29]. Typical values

[1]Physics Department, Durham University, South Road, Durham, UK. [2]Present address: London Centre for Nanotechnology, University College London, London, UK. [3]Present address: Department of Engineering Science, University of Oxford, Parks Road, Oxford, UK. ✉e-mail: william.trewby.10@ucl.ac.uk; kislon.voitchovsky@durham.ac.uk

of $D$ range from 1-10 $\mu m^2$ $s^{-1}$ for supported membranes (see Supplementary Note 1.1 for a summary of representative values). Single molecule optical techniques[30–34] can provide mobility measurements with a spatial resolution as high as 10-20 nm, but they rely on careful fluorescent labelling of the molecule of interest and are temporally limited. Further, fluorescent techniques can provide conflicting conclusions about the extent of membrane ordering[35], with the dye itself potentially influencing the measurement. Even the use of a singular value of diffusion coefficient implicitly assumes well-behaved Brownian molecular motion, in contrast with the crowded environment of cell membranes where non-Brownian (anomalous) diffusion regimes are often the norm[10,11]. In this mindset, some studies have attempted to quantify the viscoelastic properties of model lipid membranes[36], but reports of non-Newtonian behaviour remains controversial[37] with experimental limitations casting doubts on the results.

To fully characterise the local viscoelastic forces and the resulting molecular dynamics of biological membranes, the appropriate spatial and temporal scales must then be interrogated simultaneously, with mechanical measurements offering a direct handle on forces. Practically, this implies a need for nanometre spatial resolution, and microsecond temporal resolution, all while operating in-situ, locally, and on unmodified membranes. Atomic force microscopy (AFM) offers suitable spatial resolution, with the use of a nanometre-sharp probe making it possible to routinely map membranes and live cells locally, capturing sub-molecular details[5,38]. However, this is contrasted with AFM's temporal resolution, which is typically of the order of seconds. Recent advances in the field have opened the milliseconds realm[39,40], with microsecond measurements possible in the direction perpendicular to the sample[41] and dependent on a well-characterised topographical contrast. This enables detailed visualisation of the dynamics of proteins' motion, but the absence of lateral sensing precludes tracking the surrounding lipids dynamics. Here we address these limitations in the context of viscoelastic measurements by combining AFM with a bespoke fast in-plane actuator. This enables us to perform high-frequency nano-rheological measurements on model and native fluid bio-membranes in solution with timescales down to 20 $\mu$s. We acquire high-resolution images in amplitude modulation[42] and subsequently quantify the local viscoelastic forces. We show that it is possible to track the molecular dynamics within fluid membranes across three orders of magnitude in velocity while retaining the spatial resolution of traditional AFM measurements. Significantly, the approach relies on small shear amplitudes of the actuator, making the technique inherently high resolution and easy to implement on most commercial AFMs without the need for specialist equipment. Molecular dynamics (MD) simulations of the measurement process validate the experimental approach and provide further insights into the interplay between molecular mobility in the membrane and in the adjacent water layer.

## Results

### Principles of the technique
Relying on conventional AFM imaging to track molecular mobility is unrealistic for moving lipids, given their size and diffusion velocities, especially if the goal is to operate in-situ and in biologically relevant conditions. Instead, our approach adapts the AFM to effectively operate as a nano-rheometer[43–46]: a small (<7 nm) lateral shear oscillation is imposed on the sample and probed locally by the tip. The coupling between the oscillating sample and the tip is achieved through the small group of molecules located immediately under the tip apex ($10^2$–$10^3$ in the case of lipids). As a result, the shear (in-plane) forces and oscillation phase difference measured by the tip contain direct information about the molecules' viscoelastic behaviour, itself related to molecular mobility. To reach the relevant shearing velocities while maintaining nanoscale locality (small amplitudes), high shearing frequencies are needed. This is not possible with standard AFMs due to

the mass and resonance frequencies of the moving parts[46]. Instead, we make use of a small shear actuator that can fit easily on most AFM sample stages (Fig. 1a; further details are given in Supplementary Note 2), effectively augmenting the microscope with capabilities for high frequency shear measurements, while retaining its high force- and spatial-resolution. To avoid stimulating flexural or torsional resonances of the measuring cantilever and minimise hydrodynamics effects, we use small cantilevers (USC-F1.5-k0.6, Nanoworld, Switzerland) with nominal planar dimensions of $7 \times 3$ $\mu m^2$. Calculation of the cantilevers' torsional stiffness constant, $k_\Phi$, and sensitivity, $\gamma_\Phi$, were determined from the torsional thermal spectrum in air and in solution[47] (see "Methods" for details). The high torsional resonances make it easy to operate sub-resonance, but at the cost of high $k_\Phi$. The resulting sensitivity is typically in the range of a few pN of lateral force.

An example nano-rheology measurement is shown in Fig. 1b for a fluid supported lipid membrane composed of 1,2-dioleoyl-*sn*-glycero-3-phosphocholine (DOPC). During a measurement, the tip approaches the membrane from the liquid and applies an increasing normal force, compressing (yellow highlight) and eventually rupturing the membrane and becoming pinned to the underlying substrate. At a shear frequency of 1 kHz (red), the shearing velocity is much slower than the natural diffusion of the molecules, which can easily rearrange around the tip so that no shear force is transmitted. In contrast, for high frequencies of 50 kHz (black) a measurable shear force and phase lag on the tip can be observed. The explore membrane's dynamic behaviour, we systematically varied the shearing velocity by either changing the applied shear amplitude, $A_S$, or the associated frequency, $f_S$. Both parameters can be controlled independently (Fig. 1c). The shear force can be intuitively understood as the resistance of the lipids trapped under the tip to the imposed shearing motion: if the effective diffusion of the group of lipids is slower or comparable to the imposed shear velocity, a net shear force arises. Consistently, the shear force increases with $A_S$ and $f_S$ at a given load (i.e., membrane indentation). Interpreting the associated phase is less straightforward. In rheology, a phase of 0° indicates a perfectly elastic (solid-like) behaviour while 90° indicates a purely viscous (fluid) behaviour. The phase is not defined if no shear force can be measured, here before the tip-membrane contact. Over the indentation region, the phase consistently increases towards a more viscous behaviour, but it always retains an elastic component. This behaviour is still observed when varying $A_S$ and $f_S$. Upon rupture of the membrane (b), the tip eventually pins onto the substrate, reaching a phase of zero at higher loads (not shown). While this last point confirms the validity of the nano-rheology measurement, the apparent viscoelastic behaviour contradicts existing literature which suggests a purely viscous behaviour for simple fluid membrane[36,37]. In contrast, interfacial water is notoriously sluggish due to correlated interactions[48–51], offering a viscoelastic response to the shearing tip. Here, our observations can be rationalised by the AFM tip sensing the coupled dynamics of both, the lipids and the interfacial water in contact with the bilayer. As the tip indents the bilayer, the interfacial water between the tip and the membrane is progressively expelled, leading to a lipid-dominated, more viscous behaviour (Fig. 1b, c).

To test this interpretation, we conducted experiments aimed at varying the properties of the lipid membrane in a controlled manner without significantly affecting the interfacial water. We first probed the nano-rheological properties of a 1,2-dimyristoyl-*sn*-glycero-3-phosphocholine (DMPC) bilayer just above its transition temperature, progressively increasing the temperature by steps of 4 °C (Fig. 1d). The expected trend emerges[52], with the shear force decreasing as the temperature increases for a given indentation, reflecting the enhanced lipid mobility. The phase evolution also follows the trend, although with all curves overlapping within error. Alternatively, we added set amounts of cholesterol to the DOPC bilayer, thereby reducing its average lipid mobility while still ensuring a homogenous phase[53]. This

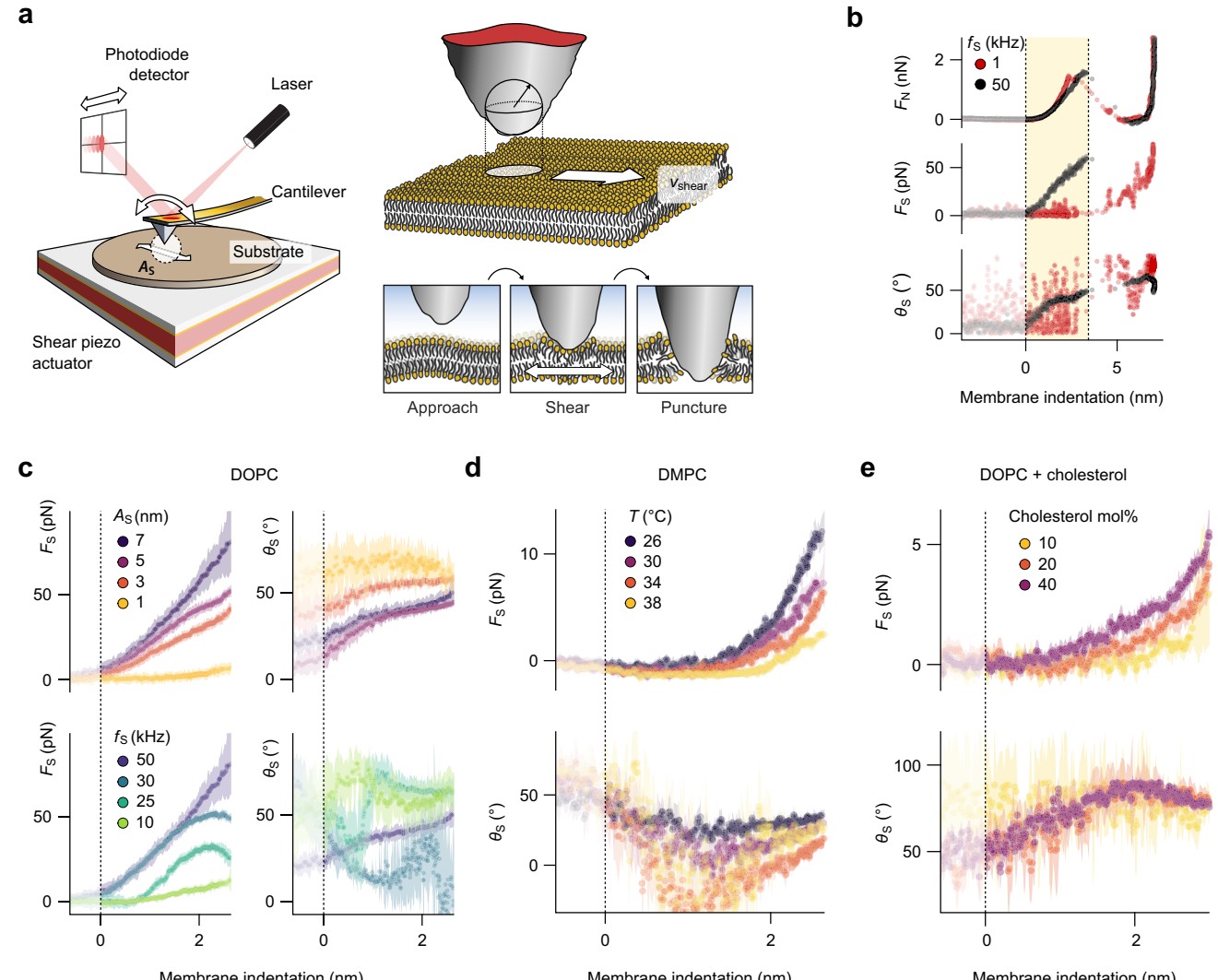

**Fig. 1 | Nanoscale tracking of molecular mobility with AFM nano-rheology.**
Schematic of apparatus (**a**) (not to scale), comprising a shear piezo actuator, mounted directly onto the AFM scanner to apply a controllable shear amplitude $A_S$. An external lock-in amplifier measures the shear force, $F_S$, and phase, $\theta_S$, experienced by the tip at a given shear frequency, $f_S$. In a spectroscopy measurement on a supported DOPC lipid membrane (**b**), the tip approaches the membrane from the bulk solution. Away from the membrane, no normal force $F_N$ is applied, and no shear force measured aside from hydrodynamic drag. We hence set $F_N = F_S = 0$ as a reference and the shear phase is undefined (semi-transparent region). As the tip begins to interact and press on the membrane (yellow highlight) $F_N$ increases, but the magnitude of $F_S$ depends on the shear velocity. At 1 kHz, $F_S \approx 0$ and $\theta_S$ is still undefined (red), but $F_S$ and $\theta_S > 0$ at 50 kHz (black), with $\theta_S$ increasing with $F_N$. Once $F_N$ exceeds a threshold, the membrane ruptures and the tip eventually becomes

pinned to the substrate underneath. (**c**) In the membrane indentation region, $F_S$ increases with both $A_S$ (for fixed $f_S$) and $f_S$ (for fixed $A_S$), indicating a consistent increase of $F_S$ with the shearing velocity $v_S$. Similarly to (**b**), $\theta_S$ tends to increase with $F_N$ in any given situation, indicating a transition from an elastic-dominated response to a more viscous-dominated behaviour of the bilayer-water system. (**d**) The sensitivity of the measurement can be tested on a pure DMPC bilayer at different temperatures above its gel-fluid transition. Qualitatively, the behaviour is as for DOPC, but the $F_S$ decreases with increasing temperature, indicating a more fluid membrane[52]. Similarly (**e**), adding cholesterol to a DOPC membrane reduces its molecular mobility[53], increasing the measured shear force. $A_S = 5$ nm in (**b**) and 7 nm in (**c**, lower panel), $f_S = 50$ kHz in (**c**, upper panel). In (**c**–**e**), the curves and shading represent the mean of $n > 15$ force curves and one standard deviation respectively. Source data are provided as a Source Data file.

last experiment is more challenging, because it requires comparing different samples, measured in different sets of experiments. The results (Fig. 1e) still exhibit the expected trend with higher shear force measured for larger concentrations of cholesterol. The phase is similar in all cases (within error) over the indentation region, qualitatively following the behaviour observed on DOPC. In all cases, it is straightforward do extract the shear and loss moduli of the sample, keeping in mind that fact that they contain information about both the lipid bilayer and the coupled interfacial water layer. Consistent with the proposed interpretation of the result, the loss (viscous) modulus captures better the expected membrane fluidity trends than the storage (elastic) modulus (see Supplementary Note 3 for details).

Generally, all the experiments qualitatively support the proposed interpretation of the nano-rheology measurements, whereby the shear force is dominated by the behaviour of the lipids at higher indentations, while the phase carries mixed information about both the interfacial water and the lipids. To move beyond qualitative considerations, independent insight is needed. We therefore conducted computer simulations of the nano-rheological measurement process.

### MD simulations
To confirm the validity of the proposed interpretation of the AFM measurements, we replicated the experiment in-silico using coarse-grained MD simulations (Fig. 2a, see also "Methods" for details). Given

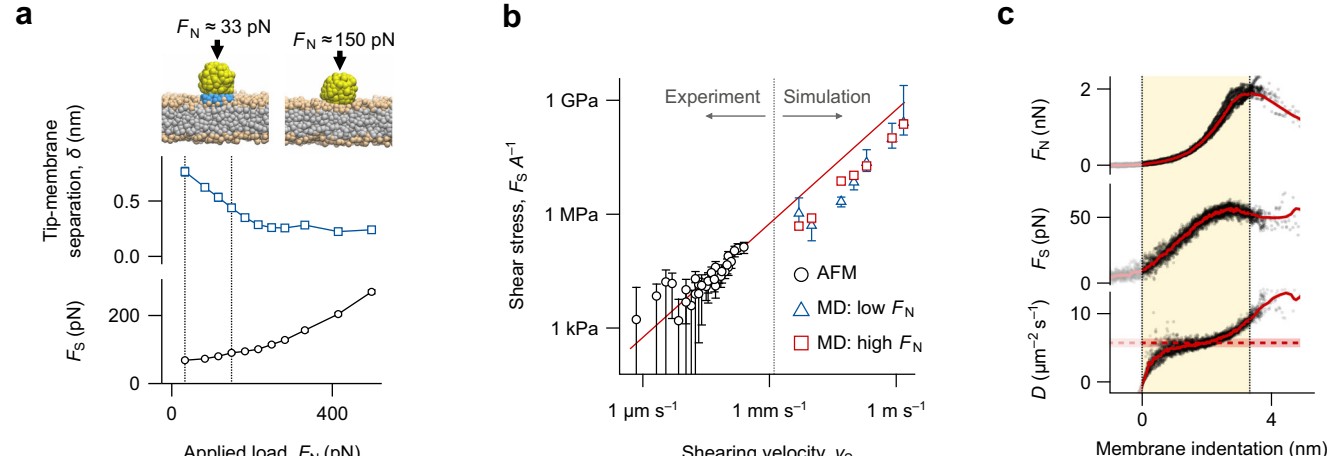

**Fig. 2 | Molecular dynamics simulations of the nano-rheological measurement.** (**a**) Coarse-grained MD simulations uncover the nanoscale interfacial behaviour as a function of $F_N$: in the low $F_N$ regime (33 pN), a layer of lubricating water molecules (blue beads) remains between the tip and the membrane, limiting $F_S$. As $F_N$ increases beyond ≈150 pN, the tip contacts the membrane and $F_S$ increases rapidly, indicative of a lipid-dominated viscous friction regime. Silica beads appear yellow; lipids' headgroup tan and their carbon chains grey. Water molecules not located between the tip and the membrane are not shown for clarity. (**b**) The simulations make it possible to directly compare in-silico and experimental measurements after normalisation of $F_S$ by the shearing area to compensate for the smaller simulated tip. The same linear behaviour is observed in both cases over six orders of magnitude (log-log scaling). The red line represents a linear fit to the AFM data (black circles), imposing passage through the origin. The imposed $F_N$ is 850 pN for AFM

and 33 pN or 88 pN for MD (blue triangles at low load, red square at high load). The relatively low force used in MD is necessary to ensure stable simulations at all velocities. (**c**) The linearity of the $F_S$ with $v_S$ suggests a simple relationship between the measured shear force, the contact area and shearing velocity. Using the Evans-Sackmann model and Einstein relation allows a direct measurement of the local diffusion coefficient, $D_{eff}$, of the lipids trapped underneath the shearing tip (Eq. (1)). This can be calculated for each point of a spectroscopy curve, here illustrated for a DOPC bilayer. Over the indentation region (yellow highlight), $D$ increases rapidly to reach a plateau at $D = 5.8 \pm 0.6 \ \mu m^2 \ s^{-1}$ (dashed red line). The shearing velocity in (**a**) is $v_S = 200 \ mm \ s^{-1}$. In (**c**), individual data from $n = 15$ force curves are overlaid in semi-transparent black, to emphasise that the calculation of D can be performed using a single point; the mean values are shown in red. Source data are provided as a Source Data file.

the size of the experimental system, some rescaling is needed: the tip is modelled as a 2 nm silica sphere shearing a DOPC bilayer in water, with the lipid headgroups of the lower leaflet immobilised to mimic the supporting substrate. The simulations highlight the significant role of the interfacial water, in particular the hydration water molecules strongly adsorbed to the lipids headgroups. These water molecules remain between the virtual tip and the membrane surface at lower applied normal forces (coloured blue in Fig. 2a, left inset), but are progressively removed at higher loads (Fig. 2a, right inset), confirming the proposed experimental interpretation. The transition from hydrated to non-hydrated shearing coincides with the appearance of a plateau in the average tip-membrane separation and an increase in the measured shear force. At low applied loads (low indentation), the shearing is dominated by the dynamics of the interfacial water adsorbed to the membrane, while the plateau coincides with the 'direct' shearing of the lipids under the AFM tip. In this second regime, the measurement is dominated by the behaviour of the lipids, consistent with the nano-rheology observations.

The present simulations also highlight the fact that water plays a significant role in controlling the dynamics and exchanges at biointerfaces: nanoscale objects and molecules moving near biomembranes encounter a protective water layer effectively acting as a lubricant[54]. Relatively higher normal forces are required to remove this lubrication layer at higher velocities, here 150 pN for a 2 nm object moving at 200 mm s$^{-1}$ (Fig. 2a).

**Towards a quantitative model**

Experimentally and for a given applied load (i.e. membrane indentation), plotting the measured shear force against the root mean-squared shearing velocity ($v_S$) reveals a linear increase of $F_S$ with $v_S$ (Fig. 2b). This remains true regardless of whether $A_S$ and $f_S$ are used to change $v_S$ (Fig. 1c). The MD simulations confirm that the AFM can directly probe the dynamics of the lipid at higher loads, but direct

comparison with the experimental data requires some normalisation of the shear force measured by the tip-lipid contact area $A$ –equivalent to the shear stress on the membrane. This strategy implicitly compensates for differences in applied load or tip size since both affect the shear force through the contact area. Figure 2b shows the stress, $F_S/A$, for the AFM measurements and the MD simulations obtained over six orders of magnitude in shear velocity, showing a good agreement. The main source of error comes from the estimate of $A$, challenging for the MD simulations due to the coarse graining of the system and the relatively small size of the tip. An incorrect estimate of $A$ can shift the datapoints in Fig. 2b vertically, but the linear dependence on velocity for both techniques confirms the interpretation of the AFM data through Eq. (1) (presented hereafter). The fact that the linearity is preserved over six orders of magnitudes is remarkable but consistent with the system always remaining in a low Reynolds number regime. The simulations also further highlight the importance of hydration waters on the interfacial shearing behaviour.

Overall, the linearity of the shear force with the shearing velocity suggests the existence of a simple relationship between the measurable parameters and the diffusivity of the molecules under the tip. Assuming that the shear force is a directly linked to the finite mobility of the lipids under the tip apex, the so-called Einstein relation[29] for diffusion can be used to extract both molecular mobility and diffusion coefficient, where the shearing velocity is analogous to the terminal drift velocity under the applied shear force. This effective diffusion coefficient must be scaled to reflect the fact that the tip, despite being nanometre-sharp, interrogates multiple molecules as it presses against the membrane. For supported membranes in solution, this is captured by the Evans-Sackmann model[55,56] where the effective diffusion coefficient $D$ of a disk-like inclusion in the membrane of area $A$ (here, the lipids trapped beneath the shearing tip) scales with the inverse of its area. Here, $A = 2\pi R_{tip}\Delta h$ with $R_{tip}$ the tip radius and $\Delta h$ its indentation depth into the membrane. Both quantities are directly

accessible through the measurement and $R_{tip}$ can be obtained separately. Combining the Einstein and Evans-Sackmann models leads to the following equation that directly links the local diffusion coefficient of lipids with experimental observables (see Supplementary Note 1.2 for details):

$$D = k_B T \frac{2\pi R_{tip} \Delta h}{A_{lipid}} \frac{f_S A_S}{\sqrt{2} F_S}, \qquad (1)$$

with $k_B$ and $T$ the Boltzmann constant and absolute temperature respectively, and $A_{lipid}$ the mean area per lipid (the diffusing specie measured) within the membrane. We emphasise the fact that, bar $A_{lipid}$ which is characterised elsewhere[27], all the terms in Eq. (1) are directly accessible as part of our experiment, with no scaling 'fudge factor', beyond the assumption of a spherical tip apex. Additionally, a single data point of a spectroscopy curve where the tip is in contact with the membrane is in principle sufficient to deduce a local diffusion coefficient, provided the membrane is not significantly distorted. To test the validity of our approach, Eq. (1) was applied to a set of 15 experimental curves obtained on the DOPC membrane, calculating a diffusion coefficient for each data point (Fig. 2c). The measurements are well reproducible over the set, with $D$ increasing rapidly as the tip contacts the membrane but showing comparatively little evolution over the indentation region. This confirms the robustness of the method, provided a suitable region of the curve is selected (highlighted in Fig. 2c). Objectively, this can be defined as the region where the derivative of $D$ with respect to the indention depth is minimal, corresponding to a robust tip-membrane contact, but without significant deformation or rupture. The obtained diffusion coefficient, $D = 5.8 \pm 0.6\,\mu m^2\,s^{-1}$ (confidence interval: 99.7%), is slightly high, but still within literature values for this lipid system (see Supplementary Note 1.1). Inevitably and as for all AFM-based techniques, there is uncertainty over the precise tip geometry. However, the compensating relationship between tip radius and indentation depth for any given load means the data are robust to changes in $R_{tip}$ (see Supplementary Note 4, and ref. 57). This is another advantage of the approach, with tip size impacting mainly on lateral resolution but with limited effect on the derived diffusion values.

## Applications to ternary model membranes

Having established the robustness of the method, we illustrate its capability on complex model membranes comprising a ternary lipid mixture of sphingomyelin (SM), DOPC and cholesterol, at a respective molar ratio of 4:4:2. This membrane exhibits distinct liquid-ordered ($L_O$) and liquid-disordered ($L_D$) domains easily identifiable from their height difference (Fig. 3a), as expected from the mixture's high critical temperature[58]. Mapping of the associated local diffusion shows the expected lower diffusion coefficient in the $L_O$ regions (Fig. 3b). The associated spectroscopy curves obtained over $L_O$ and $L_D$ regions (Fig. 3c) overlap over the first ≈1.5 nm as the calculated $D$ increases with indentation, followed by the domain-specific plateau used to derive the diffusion coefficient in each location. The overlap region of the spectroscopy curves is consistent with the lipids' hydration water acting as a lubricating layer[59,60], as observed in our simulations (Fig. 2a). Here, the exposed lipid headgroups are identical on all regions, explaining the apparent similarity of the curves. Statistical analysis of $D$ over the plateaus yields histograms with a spread of ≈20 % around the average, especially for the $L_O$ regions (Fig. 3d). This spread represents genuine nanoscale variations of the diffusion across a given domain and not an experimental error. The spread is exacerbated over the $L_O$ domains due to the lower molecular mobility rendering the measurement more sensitive to local molecular variations. This dependence on local variations suggests some possible spatial correlations in the diffusivity across the membrane. To test this hypothesis, we calculate the so-called spatial lag[61], $Y_{SL}$, across the entire diffusivity

map. $Y_{SL}$ describes the statistical correlation between the normalised diffusion coefficient of adjacent measurement locations within each lipid phase (Fig. 2e, see also Supplementary Note 1.3 for further details). The statistical test calculates a correlation value (Moran's value[61], $I$) from the different $Y_{SL}$; $I > 0$ indicates spatial correlations. Here we find a positive correlation, with $I = 0.158 \pm 0.042$ ($n = 816$), as illustrated graphically in Fig. 2f, and with a strong statistical significance ($p < 0.0025$). This confirms that the local diffusion is globally dependent on the immediate nanoscale environment (see Supplementary Note 1.3 for more details on the test). Conducting the same analysis separately over the $L_O$ and $L_D$ domains confirm that spatial correlations in the diffusion exist over both domains, but with degree of correlation three times higher for $L_O$ regions ($I_{LO} = 0.146 \pm 0.007$, $n_{LO} = 291$; $I_{LD} = 0.051 \pm 0.049$, $n_{LO} = 461$). The results are statistically significant in both cases, but the fragmented geometry of the domains and the reduced data subset available for each region renders the analysis less robust than over the entire image. The boundary regions may also play an important role in setting lateral correlations, although not easily addressable here due to the small size of the relevant region. Overall, the results confirm that the spread in diffusion values over the different domains (Fig. 3d) reflects some spatial correlations within the membrane. They also establish the need for local, nanoscale measurements of molecular mobilities to derive meaningful data: if spatial correlations are clear in idealised ternary bilayers on an atomically flat substrate, they are bound to be more significant in complex natural membranes.

## Measurements on native bovine eye-lens membranes

Building on the experimental and computation results obtained on model membranes, we performed nano-rheological measurements on fragments of native bovine eye lens membrane[62,63] (Fig. 4). Lens membranes represent an interesting system to interrogate with our technique due to the high cholesterol content and the existence of regions of heterogeneous dynamic behaviour over distances of tens of nanometres. This includes more fluid, lipid-rich domains (Fig. 4a, highlighting arrow), protein-rich domains—highlighted by white dashes—with natural tetrameric aquaporin crystals (highlighting arrow), and intermediate regions where protein crystals are assembling/disassembling (highlighting arrow). These intermediate regions have significant functional relevance in the development of lens cells, enabling the formation of suitable protein channel arrangement to connect with adjacent cells and ensure vital microcirculation of metabolites. Shear measurements taken over these different regions of the same membrane fragment evidence important variations in the local diffusivity. Over 150 nm, $D$ varies by more than an order of magnitude, with distinct variation over distances as small as 20 nm. It is important to note that the diffusion coefficients in Fig. 4d were derived with Eq. (1), implicitly assuming a fluid membrane and the lipids as the main diffusing specie probed. This is not necessarily true over protein-rich regions (positions 6, 7), where the area of a protein should be used instead of that of a lipid molecule in Eq. (1). This substitution would further reduce the measured diffusion coefficient by an order of magnitude, consistent with the observed immobile protein assemblies. Over protein crystals, the assumption of a fluid membrane no longer applies, and Eq. (1) ceases to be valid. The tip can still be used to disrupt the protein assemblies, reversing the ordered domains in regions with higher molecular mobility, leading to the recovery of a finite diffusion coefficient (Supplementary Note 5). However, it is then no longer clear which molecular diffusion (proteins or lipids) is being probed. Taken together these results support the idea of short-ranged molecular gradients within the membranes modulating the local membrane biophysical properties and controlling protein self-assembly, consistent with the concept of rafts[19] locally and transiently modifying the membrane's properties. This illustrates both the versatility and precision of the method, as well as the need for

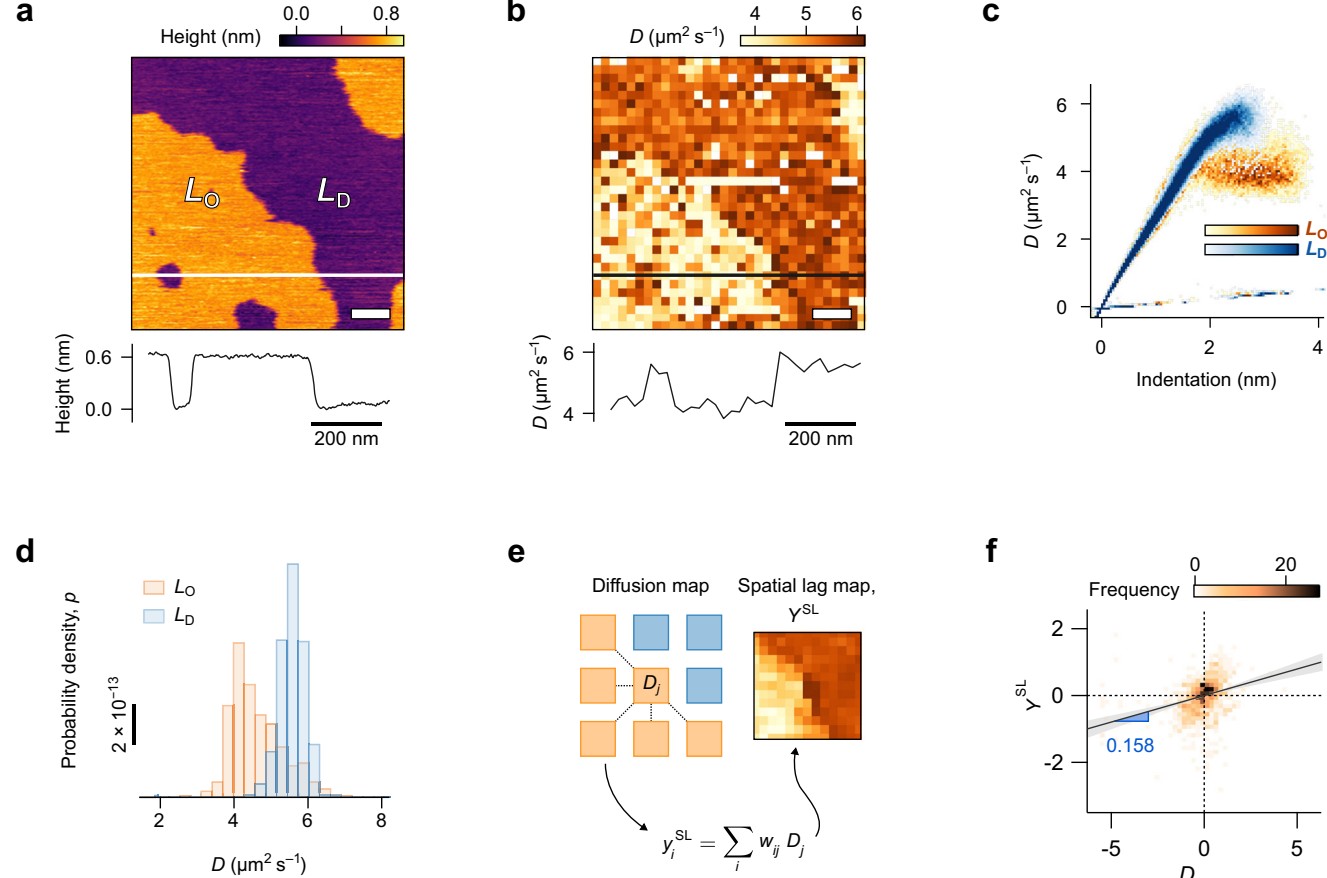

**Fig. 3 | Quantification of the local diffusion coefficient in a complex model membrane. a** In the ternary bilayer containing DOPC, sphingomyelin and cholesterol, $L_O$ domains appear higher than the $L_D$ domains in topography due to differences in lipid packing. Both phases are fluid, but systematic mapping of the diffusion coefficient (**b**) shows the $L_D$ regions to exhibit higher diffusion values. (**c**) Plotting the entire dataset of spectroscopy curves over both phases shows an identical behaviour for indentations <2 nm due to the tip probing primarily the hydration layer of the membrane. Beyond this point, the diffusion values diverge to a clear bimodal distribution ascribable to the two types of domains. This is best illustrated by a histogram of the diffusion coefficients derived over each domain (**d**). **e** Spatial correlations of the measured diffusivity can be observed within each type of domain, illustrated here for the $L_D$ regions. Correlations between adjacent diffusion values of a given phase can be quantified using the so-called spatial lag, $Y_{SL}$, representing the weighted average of the diffusion coefficients directly connected to the location considered ($w_{ij}$ being the proximity weighting element). This method makes it possible to statistically test possible spatial correlations (**f**) through the so-called Moran value[61] (see Supplementary Note 1.3 for details). The test reveals clear lateral correlations, illustrated by the non-symmetrical distribution of $Y_{SL}$ values around the origin (**f**) with a Moran value of $I = 0.158 \pm 0.042$. This indicates a hight statistical confidence in the presence of spatial correlation ($p = 2.25 \times 10^{-3}$) when compared to an uncorrelated system. Scale bars represent 100 nm in (**a, b**), with $f_S = 25$ kHz and $A_S = 5$ nm. Diffusion values for the map (**b**) and histogram (**d**) were computed from indentations of $\Delta h = 2.0 \pm 0.5$ nm. Determination of the curves as $L_O$ or $L_D$ was done using the height information (**a**) taken immediately before and after the measurement (see Supplementary Fig. 11 for details). Colour scales in (**c**) represent data point frequencies of 0–50 ($L_O$, red scale) and 0-100 ($L_D$, blue scale). Source data are provided as a Source Data file (except for (**c**) available as raw data from the online repository).

such measurements to be routinely applied to biological membranes if to derive a functional picture.

## Discussion

We have developed a novel approach to quantify the in-plane molecular mobility of supported membranes with nanometre spatial resolution. Our technique uses nano-rheology to surpass the conventional temporal limitations of AFM to resolve in-plane dynamics and capture the local membrane dynamics. The approach offers some advantages over existing techniques, but it also has its limitations. The main advantages stem from the 'mechanical' nature of the measurement which provides a full viscoelasticity picture of the interfaces, probing both the lipids and the interfacial water. This measurement effectively mimics what a nanoscale object or molecule diffusing in the membrane's vicinity experiences, probing the forces at play. It operates in-situ, locally, at the nanoscale and does not require any chemical or physical labelling of the sample. The ability to quantify the membrane's

viscoelastic response may be particularly useful for complex systems with significant local variations in mechanical properties and dynamics, something challenging with existing approaches. The technique's limitations primarily come from the need for physical access to sample which must be immobilised. In biological systems, this limits measurement to outer membranes complexes if to be carried out on live cells, or to supported fragments of cells and organelles, as in this paper. Beyond the measurement itself, its interpretation of the results in terms of local membrane mobility hinges on assumptions that must be verified if to be valid. First, the sample needs to be fluid. Second, the Evans-Sackman model[55,56] used in the model developed here is only valid for supported lipid membranes. A more general Saffman-Delbrück relationship exists for suspended membranes[64], but it is less favourable for achieving highly localised measurements due to logarithmic dependence on the area probed. Third, our model assumes knowledge of the diffusing molecules species being probed and their respective size, as described in Eq. (1). While straightforward

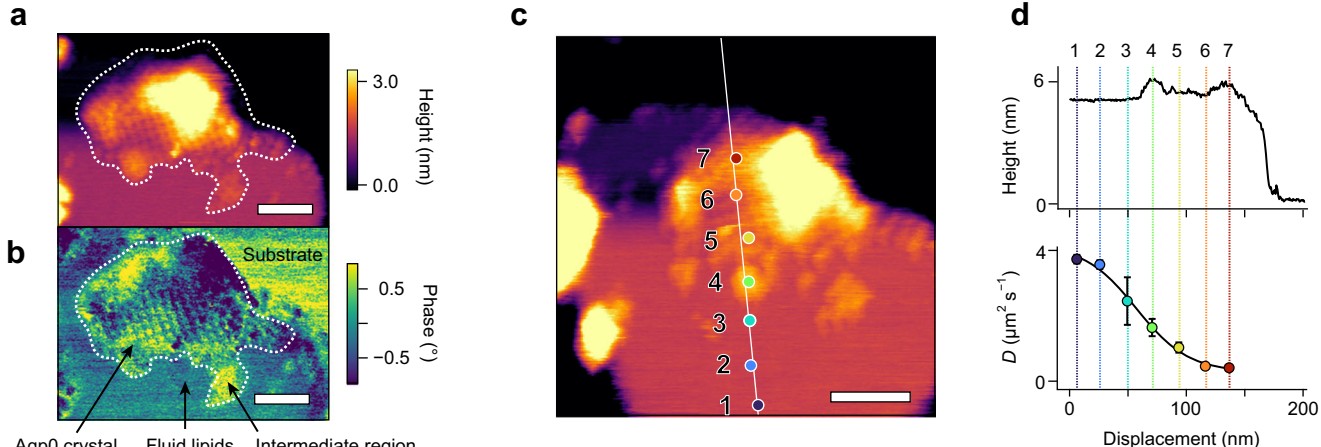

**Fig. 4 | Nanoscale diffusivity mapping of a native bovine eye lens membrane.** High resolution amplitude modulation AFM imaging of a membrane patch (**a**) reveals regions with a nanoscale AQP0 tetrameric crystal (dashed outline) surrounded by a fluid lipidic region and intermediate regions where protein crystal assembly/disassembly occurs. The separation of these functional domains is clear from the topography (**a**), also highlighted by the tip's oscillation phase image (**b**). Nano-rheological measurements conducted at specific locations across the sample (**c**) reveal important variations on a 20 nm scale across the membrane (**d**). This is consistent with local changes in the molecular dynamics of the lipids across fluid and assembling regions, emphasised by the fitted sigmoid of the diffusion coefficient (solid line, (**d**)). The error bars in (**d**) represent the standard deviation on the number of measurements with $n = 8$ (1), $n = 5$ (2), $n = 5$ (2), $n = 5$ (3), $n = 7$ (4), $n = 7$ (5), $n = 7$ (6) and $n = 5$ (7). Scale bars in (**a**, **b**) and (**c**) represent 40 nm and the colour scales cover 3.5 nm (**a**, **c**) and 1.75° (**b**). Source data are provided as a Source Data file.

for lipid membranes, this becomes more challenging for plasma membranes which contain a high fraction of proteins and sugars. Finally, the derived diffusion coefficient values carry an error linked to the imperfect geometry of the system, compared to the idealised model: the tip apex is never perfectly spherical, and its measured size is to be taken as effective. However, the resulting error is systematic and affects each measurement with a given tip in the same way. The strength of the technique therefore lies in mapping relative variations across a sample, as illustrated in Figs. 3, 4. On model biological membranes, the results quantify local variations in the lipid diffusivity even within apparently homogenous phases, revealing nanometre-scale correlation effects. Measurements on a native lens membrane also illustrate the potential of the technique to map functionally relevant nanoscale changes in molecular diffusivity across complex membranes. The ability of the technique to measure the viscoelastic response of the full system (membrane and interfacial water) shows that while the system is viscoelastic, this is primarily driven by the interfacial water with simple lipid membrane behaving mainly as viscous, hence enabling Eq. (1) to be applied. This emphasises the interplay between the dynamics of the lipids and that of the water, as also clear from the interfacial storage (elastic) and loss (viscous) moduli (Supplementary Note 3) and confirmed by the computer simulations.

In terms of spatiotemporal resolution, the technique is ideally placed to complement existing optical methods[32–34] based on super-resolution and able to quantify the diffusion of labelled species in-vivo with typically ≈1 ms and 50-100 nm resolution. For scanning probe based methods, we anticipate the technique to be particularly useful for quantifying the diffusion of lipids within fluid membranes, complementing existing approaches[41] developed to track the conformational changes of proteins on a similar timescale.

## Methods
### Assembly of the nano-rheological device
The device (see Supplementary fig. 2) consisted of a low-voltage shear piezo actuator with copper contacts and alumina-based end plates (PL5FBP3, Thorlabs, NJ, USA), which has a nominal resonance frequency of 1.9 MHz. This was affixed with two-component epoxy (Araldite Ultra-Strong, Basel, Switzerland) to a standard steel AFM puck (15 mm diameter, Agar Scientific, UK) on one side for thermo-

mechanical coupling to the AFM stage. The piezo, puck and wires were then covered with adhesive polyimide tape (Kapton, DuPont) to protect the components from the electrolytic buffer. The mica disc used as a substrate (3 mm diameter, V-1 quality, SPI supplies, PA, USA) was then affixed to this using epoxy. Finally, to ensure adequate thermal conductivity between the AFM's Peltier stage and the sample fluid droplet, conductive silver paint (Leitsilber, Ted Pella Inc., CA, USA) was applied from the steel puck to the vicinity of the mica (≈2 mm away). Details of the device calibration, testing and optimisation are presented in Supplementary Note 2. The entire device was installed in a commercial AFM (Asylum Research Cypher-ES, Oxford Instruments, Oxford, UK) and connected to an external lock-in amplifier (MFLI, Zurich Instruments, Zurich, Switzerland).

### Sample preparation
All glassware, including the lipid mini-extruder (Avanti Polar Lipids, Birmingham, AL, USA) components, were cleaned by progressive sonication and rinsing in 5 vol.% Decon-90 (Fisher Scientific, Loughborough, UK); ultrapure water (18.2 MΩ cm, Milli-Q, Merck-Millipore, Watford, UK); IPA (2-propanol, > 99.5%, Fisher Scientific, Loughborough, UK), followed by a final rinse in ultrapure water, before drying either under a nitrogen stream or at 60 °C. In all cases, the imaging and deposition solution (hereafter, "the buffer") was composed of 150 mM NaCl and 2 mM CaCl$_2$ (both Sigma Aldrich, ACS reagent, > 99% purity (NaCl) and > 96% purity (CaCl$_2$)) in ultrapure water. It was made at 10× concentration, syringe filtered and diluted to 1× with ultrapure water as required.

The purified supported lipid bilayers were produced from small, unilamellar vesicles (SUVs) followed by the vesicle fusion method[65]. 1,2-dioleoyl-*sn*-glycero-3-phosphocholine (DOPC), sphingomyelin (SM, porcine brain) and cholesterol were purchased from Merck/Sigma-Aldrich (Gillingham, UK) and used without further purification. The appropriate lipid molar ratios (either 100% DOPC, 100% DMPC, 9:1 DOPC:cholesterol, 8:2 DOPC:cholesterol, 6:4 DOPC:cholesterol, or 4:4:2 DOPC:SM:cholesterol) were combined in chloroform and the solvent then gently evaporated under a flow of nitrogen followed by > 2 h in a vacuum desiccator to ensure complete evaporation of the solvent. The lipids were then rehydrated in the imaging buffer solution (150 mM NaCl, 2 mM CaCl$_2$) to a concentration of 1.0 mg mL$^{-1}$. The

samples were first homogenised by vortexing for 10 s, and then bath-sonicated at 40 °C for 30 min to form multi-lamellar vesicles. 1.2 mg of this solution was equilibrated in a mini lipid extruder (Avanti Polar Lipids, Birmingham, AL, USA) at 50 °C for 20 min, before extruding 27 times through a 100 nm filter (Whatman, Sigma Aldrich) to produce a clear suspension of SUVs that was kept at 5 °C for no longer than 5 days. On the day of an experiment, the shearing device was installed in the AFM and the stage heated to 45 °C. The mica was then cleaved and rinsed with $3 \times 5 \mu L$ of the imaging buffer. $5 \mu L$ of the SUV suspension was pipetted onto the mica, which was incubated at 45 °C for 10 min, before ramping down to 20 °C over 15 min. To remove any remaining vesicles, either adsorbed or in solution, the droplet was then rinsed with $15 \times 5 \mu L$ of the imaging buffer. At all times, care was taken not to let the droplet evaporate or otherwise expose the mica surface to air.

For the measurement on DMPC, the sample was brought the desired temperature using the AFM temperature control (1 °C min⁻¹) and the experiments conducted sequentially at 26, 30, 34, and 38 °C. In each case, the sample was left 5 min to equilibrate after reaching the desired temperature and silver contacts to the substrate ensured optimised thermal conduction.

For the DOPC with cholesterol experiments, the measurements were purposefully conducted in random order (10% cholesterol, 40% cholesterol, 20% cholesterol) to avoid any possible experimental bias. A same (new tip) was used for the whole experiment.

The bovine eye lens membrane fragments were prepared using a standard sequential extraction method[66,67]. In short, fresh bovine lens were decapsulated and subsequently stirred on ice for 20–30 min at a 1:2 weight-to-volume ratio of extraction buffer (10 mM sodium phosphate pH 7.4; 150 mM NaCl, 5 mM EDTA). The lens material was poured off leaving behind the residual lens. Over longer stirring times the lens became completely dissociated, and the separated lens material Dounce homogenised. The membrane-enriched fraction was then separated from the soluble protein fraction by centrifugation (Beckman JA20 rotor; 48400× g at 41 °C for 20 min.). The pellet containing the lens membranes was again Dounce homogenised to resuspend the membranes and extracted again with the same buffer. The process of resuspension, stirring on ice and centrifugation was repeated consecutively in the following buffers: 10 mM Sodium phosphate pH 7.4, 1.5 M KCl, 5 mM EDTA; 10 mM ammonium bicarbonate, 1 mM EDTA; 10 mM sodium phosphate pH 7.4, 8 M urea, and 5 mM EDTA; 0.1 M sodium hydroxide. Afterwards, the lens membranes were washed once more in extraction buffer and then resuspended in the same containing 0.01% (w/v) sodium azide, aliquoted and stored at 4 °C or frozen at −20 °C for later use. To prepare the AFM samples, an aliquot was allowed to defrost at room temperature and then mixed gently with a pipette. 20 μL was withdrawn and mixed with 60 μL of 40 mM CaCl₂. The mica surface was then cleaved and rinsed with $3 \times 5 \mu L$ 40 mM CaCl₂, before pipetting 5 μL of the lens fragment solution onto it and incubating at room temperature for 15 min.

## AFM measurements

A Cypher-ES AFM (Asylum Research, Oxford Instruments, CA, USA) equipped with a Peltier stage for temperature control ($\Delta T = \pm 0.1$ °C), photothermal excitation and a small laser detection spot was used in all cases. Small cantilevers were used (USC-F1.5-k0.6, Nanoworld, Switzerland) with a new lever used for each set of experiment. Prior to experiments, the cantilevers were rinsed consecutively with acetone, IPA and ultrapure water. Their flexural stiffnesses were calculated from their thermal spectra recorded in the imaging solution[68] and did not significantly deviate from their nominal value (typical range of 0.45-0.8 nN nm⁻¹). The levers' lateral stiffnesses and optical sensitivities were calibrated in situ using non-destructive methods based on the torsional thermal spectra[47,69] The invOLS was taken from a linear fit of the lever deflection versus z-piezo extension

when in hard contact with the mica substrate, using the Asylum Research AFM control software for Igor Pro (Wavemetrics, OR, USA).

Imaging was conducted in amplitude modulation mode, with the lever photothermally excited near its resonance frequency ($\approx 900$ kHz) such that it has a free (flexural) amplitude in the bulk imaging buffer of $A_0 \approx 1$-2 nm. The setpoint amplitude used for imaging feedback was kept $> 0.7 A_0$ to ensure gentle imaging while still accurately tracking the topography[70,71].

The spectroscopy data presented in Figs. 1, 2, 4 was collected in static force mode, with an approach velocity of 30 nm s⁻¹. The applied normal force is derived directly from the flexural deflection of the cantilever. Simultaneously, the lock-in amplifier applied a sinusoidal potential to the shear piezo, actuating it with frequencies < 50 kHz and with displacements of 1-7 nm (details of the calibration of the open-loop piezo in Supplementary Note 2). The lateral motion of the AFM laser on the photodiode (Fig. 1a) was sent into the lock-in and used to quantify the amplitude and phase offset, relative to the piezo motion. The measured amplitude was used to calculate the lateral force on the tip, $F_L$ as follows[69]:

$$F_L = k_\phi \frac{L}{(L - \Delta L)h_{tip}^2} \cdot \gamma_{tors} \cdot \Delta V_L = k_L \cdot \gamma_{tors} \cdot \Delta V_L, \tag{2}$$

where $\gamma_{tors}$ is the torsional optical sensitivity (in rad $V^1$), $k_\phi$ is the torsional stiffness of the lever (in N rad⁻¹), which is then converted to the lateral stiffness of the cantilever, $k_L$ (in N m⁻¹), by considering the cantilever's length $L$, and the height, $h_{tip}$, and setback of the tip, $\Delta L$. The torsional optical sensitivity is not intrinsic to the cantilever and depends on the particular optical path of the detection laser, as well as the AFM geometry and so must be calculated for each lever in situ from the torsional thermal spectrum[47,69]

$$\gamma_{tors} = h_{tip} \sqrt{\frac{2k_B T}{\pi k_\phi f_0 P_{DC} Q}}, \tag{3}$$

where $k_B$ is the Boltzmann constant and $T$ is the absolute temperature. The variables $f_0$, $P_{DC}$ and $Q$ are taken from a Lorentzian fit performed on the torsional power spectral density. For $k_\phi$, we make use of a method based on the levers' torsional thermal spectrum measured in both air and buffer[47].

For spectroscopy measurement at higher frequencies ($\geq 25$ kHz), a lock-in time constant of $\tau = 199.8$ μs was used, but for lower frequencies, this was progressively increased to maximise the signal to noise ratio, ending with $\tau = 1.973$ ms at a shear frequency of 1 kHz. All data points in Figs. 1c–e, 2c, and 4c represent the mean, with error bars spanning one standard deviation of >15 force curves for each parameter set.

For the diffusion map in Fig. 3b, the 'force map' mode of the Asylum Research software was used, with vertical tip velocities of 120 nm s⁻¹ and a data sampling rate of 6.25 kHz. The shearing piezo was driven with a 25 kHz signal and maximum displacement of 5 nm. The map represents the mean diffusion coefficient of the values at an indentation of $\delta = 2.0 \pm 0.5$ nm. All analysis was performed in Igor Pro (Wavemetrics, OR, USA) using custom scripts. Standard AFM images were acquired immediately before and after the map acquisition to evaluate possible drift and ensure correct interpretation of the data (more details in Supplementary fig. 11).

## MD simulations

Coarse-grained molecular dynamics (CGMD) simulations of the nano-rheology measurements were performed using the MARTINI forcefield[72] with timesteps of 40 fs. Prior to the main simulation, energy minimization was conducted, with an equilibrium simulation of 80 ns. Water and ions and the rest of the system were thermostated independently at a temperature of 300 K, deploying two Bussi

thermostats to avoid the creation of hot and cold spots in the simulation box[73]. The extended ensemble pressure coupling of Parrinello-Rahman was adapted to control the pressure at 1 bar with the $x$ and $y$ dimensions of the simulation box remaining the same, while the $z$ dimension adapts to the pressure changes independently[74]. Steered MD simulations[75] (SMD) were deployed to mimic the AFM shearing experiments as faithfully as possible. To this end, a lipid bilayer made of 722 DOPC lipids was constructed and solvated in 16000 martini water beads, (10% antifreeze beads), with a NaCl concentration of 150 mM (Supplementary fig. 12). A spherical tip made of MARTINI silica beads was constructed with a radius of 2 nm. The total size of the simulation box was $15.51 \times 15.51 \times 12.80$ nm$^3$ (see Supplementary Table 2 for details on the simulations parameters). Forces perpendicular to the membrane plane (equivalent to the normal load, $F_N$, applied experimentally) was applied to the tip while in the parallel direction the tip was pulled with a specified velocity in the range of $5 \times 10^3 - 1.5 \times 10^6$ µm s$^{-1}$.

To mimic the effect of the hydrophilic mica support on the lipids, the lipid headgroups of the lower leaflet were fixed. This was compared to simulations without any constraints (free standing). At low $F_N$, interfacial water was present between the tip and the lipids in both cases and the measured shear forces were comparable. At higher loads, this interfacial water was squeezed out for the constrained membrane only, leading to a sharp increase in shear forces (Supplementary Fig. 13). Error analysis shows that at least 30 repeats of any simulation is needed to achieve 20% accuracy on the mean. Accordingly, each pulling simulation was repeated at least 50 times. Python and bash codes were written to analyse the pulling simulation results. In all the simulations the timestep of 40 fs was adopted. As the dynamics of the system with the martini coarse graining are increased by a factor of four[76], the simulation time and the pulling speeds reported in the current study are based on the estimated real time, which is four times the simulation time. Specific parameters for the simulations are detailed in Supplementary Table 2.

## Data availability

The data that support the findings of this study are available from Durham University's Research Data Repository[77] and from the corresponding authors upon request. Source data are provided with this paper.

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

## Acknowledgements

The authors acknowledge funding from the UK Engineering and Physical Sciences Research Council (EPSRC grant EP/S028234/1) and Durham Physics (WJT). The authors are grateful to Leon Bowen for help with SEM

analysis of the cantilevers and Ruth McTiernan for help with preparation of the AQP0 samples.

## Author contributions

K.V. and W.T. designed the study. W.T. conducted the experimental measurements and analysed the data with input from K.V. M.T. conducted the computer simulations and analysed the data with input from K.V. and W.T. All authors wrote the paper and commented on the results.

## Competing interests

The authors declare no competing interests
