## [Transparent Peer Review file · Nature Communications]

Local mapping of the nanoscale viscoelastic properties of fluid membranes by AFM nanorheology

Corresponding Author: Professor Kislou Voitchovsky

Version 0:

Reviewer comments:

Reviewer #1

(Remarks to the Author)

The manuscript "Local mapping of spatial correlations in the nanoscale diffusion dynamics of fluid biomembranes" by William J. Trewby, Mahdi Tavakol, and Kislou Voitchovsky present AFM-based local mapping of the fluidity of lipid bilayers at the nanoscale using an AFM-based approach.

The paper is interesting and well written, and I am in favor of publication.

Major concern: From what I read, I think the method is very powerful to characterize the nanoscale diffusion properties of lipids in a lipid membrane (or a biomembrane, but still only the lipids. I think simplicity the author think that too, as they say when they shear over a <7nm area/line, then they measure about 100 molecules (line 83). If they were to talk about membrane proteins that number would be 1. Thus, in very general terms, I dislike the use of the words biomembrane, as it implicates something much more complex with ~50% proteins packed and sugars, etc. Obviously, when the authors would land by coincidence (likely ~50% of the time in a real plasma membrane) on a 7nm (the typical size of a membrane protein) membrane protein, then they would probe the diffusion properties of that single molecule, which would be about an order of magnitude slower than the lipids. If it's a bigger protein, ever more so. Given that the method also relies on a 'contact mode' type of tip-sample interaction, a membrane protein would likely be plugged down on the mica surface at ~50pN constant force, and the diffusion would be essentially zero.

In this context, I do not understand what the authors measure on the AQP0 patches in the eye lens membranes (cite Buzhysnky EMBO R 2007, and, Colom Nat Comm 2013, which both used AFM to image AQP0 patches in native eye lens membranes, the latter on cells). Indeed, as these authors could simply image the AQP0 patches and the authors here do that too, the diffusion of these large protein patches is essentially zero. Thus, what is the signal measured on these patches. I doubt that the presented shear method can provide a meaningful value for the diffusion of such systems.

Related to this, as the authors cite reference 38 (line 64), I feel that paper could receive a bit more consideration as the complementary method (interestingly with the same temporal ~20μs resolution) to what is presented here. Indeed the topography based (tapping mode feedback operated method) by Heath Nat Comm 2018, is perfectly suited to characterize membrane protein dynamics. On the other hand, it cannot assess lipid dynamics, which the method presented by the authors clearly can.

Thus, I recommend to rephrase the entire paper to make clear, as the methodology figure 1 shows, that the technique is mainly/only powerful to characterize the nanoscale diffusion dynamics of lipids, but that it has its limitations when it comes to larger objects such as membrane proteins.

Minor concern: The method is based on a shear measurement, and the cantilevers have typically tremendously high torsional spring constants (for shear measurements). I find almost no or absence of information in the main text, and think some the cantilever (torsional) characterizations should be moved from supplementary information to main text. Also, discussing openly the torsional stiffness, and the detection limitations to the faint torsional deflections would be a gain to the manuscript, even if it points to some limitations.

In summary I congratulate the authors to this development, and think it is most valuable. I do not request any additional experiments, but expect a thorough work-over of the text to focus and discuss what the technique is powerful for and where its limitations are.

Reviewer #2

(Remarks to the Author)

The manuscript reports an AFM-based study to measure the diffusion coefficient of lipids in supported lipid bilayer membranes. The manuscript presents a method based on the measurement of oscillatory shear forces. The experimental results are interpreted in terms of theory and coarse-grained MD simulations. The results illustrate the role of hydration water on the interfacial shearing behaviour. They illustrate also the potential of the technique to map local transitions in diffusivity in membranes.

This manuscript presents an excellent, timely and original method to measure the mobility of lipids at the nanoscale. It has the quality and relevance to be published in Nature Communications. Some revision is recommended.

Comments

1 The AFM method would always measure shear forces. Therefore equation 1 would always provide a diffusion coefficient even on a solid surface without any diffusion species. The authors should clearly explain the conditions required to use properly eqn 1 for obtaining a diffusion coefficient.

2 AFM imaging was performed in amplitude modulation AFM. The authors might consider to include some general references to this popular AFM mode (for example, R. Garcia, R. Perez, Surf. Sci. 47, 197 (2002)).

Reviewer #3

(Remarks to the Author)

The article by Trewby et al. describe the construction of a new atomic force microscopy (AFM)-based method to measure diffusion coefficients in lipid membranes at high spatial resolution. After explaining the principle of the instrument and its characterization, the authors apply it to bilayers and a biological membrane. The article is well written, the work of good quality, and the conclusions are supported by the data. However, I think the article would fit better in a more specialized, possibly technical journal as I do not see the wider interest the way it is presented currently. The technique cannot be applied *in vivo*, and the biological membrane used is a reconstitution. The technique is not specific as it measures the diffusion coefficient of all membrane components simultaneously and cannot differentiate between different membrane embedded molecules. Therefore, while it allows measuring diffusion coefficients at high spatial and temporal resolution it cannot differentiate between different components. This is different from other techniques that offer the same spatiotemporal resolution but are also specific.

Detailed comments

Line 58: There are articles which have shown to be able to measure at high spatiotemporal scales (see Eggeling et al. Nature 2009, Xiang et al. Nature Methods 2020; Sankaran et al. Nature Communications 2021). These methods could be compared to the current method in terms of spatiotemporal resolution, specificity, and usable sample types. Especially single particle tracking could be discussed as that allows measuring of a single molecule at similar spatial resolution.

What is the actual tip radius used (2nm similar to the simulated spheres?) and how much do the results depend on this factor, i.e. do the authors experimentally recover the same mean diffusion coefficient and standard deviation when the tip radius varies?

A very interesting result is that the authors see a bimodal distribution of the diffusion coefficient in Lo domains. Is there an explanation for that and does this mean the Lo domains are not homogeneous but are rather a mixture of Lo and Ld domains? The authors might also want to compare their diffusion coefficients in Lo domains to what has been measured before as the results seem to be quite high (see next point).

Line 151 and Fig. 1: The diffusion coefficient for DOPC is on the high side. It exceeds six of the cited DOPC results significantly and is only smaller than one exceptional measurement that determines a diffusion coefficient of 8 $\mu\text{m}^2/\text{s}$, i.e. at least a factor of two higher compared to all other measurements. So it is difficult to argue that the authors' results lie within the bulk of the measurements. And measurements on membranes of ternary mixtures but also on biological membranes are quite high, although they show more variability. This raises the question of how well diffusion coefficients can be measured and whether there is a bias. It would be interesting to see measurements on multiple single component bilayers and perhaps at different temperature, gel and liquid phases, to demonstrate the diffusion coefficient is unbiased and show the resolution the technique can achieve. These measurements could also be used to demonstrate that the local correlations in diffusion coefficients are reliable and follow the expected trends in liquid and gel phases.

Fig. 3d: The central data points seem to fall on an ellipse that is not centered at the origin and that has a much stronger inclination than the fitted 0.158 gradient indicated. I assume the expected correlation cloud should be symmetrical to the origin in respect to the diffusion coefficient? And what does the stronger inclination imply? That diffusion coefficients are strongly correlated? And what are the differences in these correlations within the Lo and the Ld phase? Here, too, it would be very interesting to see data on more samples, especially more ordered or gel phases, to show that the correlation length reflect bilayer properties.

Version 1:

Reviewer comments:

Reviewer #1

(Remarks to the Author)

The revised version of the manuscript "Local mapping of the nanoscale viscoelastic properties of fluid membranes: molecular mobility and spatial correlation" by Trewby et al. was carefully revised and is ready for publication. I thank the authors for their detailed and thoughtful point-by-point response, and congratulate them for their beautiful work/development.

Reviewer #2

(Remarks to the Author)

The authors have addressed satisfactorily my comments. The revised manuscript is suitable for publication.

Reviewer #3

(Remarks to the Author)

The authors provided a careful revision with new data which answers my queries. I have no further comments.

Response to reviewers

We thank the reviewers for their time, their insightful comments and suggestions which have helped us significantly improve the manuscript. In light of the significant changes required, we have conducted several sets of additional experiments to validate the method and its interpretation. We have also re-analysed some of our existing results and made extensive changes to the manuscript and its figures, including the addition of a new 'Discussion' section.

We address in detail their questions and comments point-by-point hereafter. For the sake of clarity, the questions are reproduced **in black**, our answers appear **in blue** and any text quoted from the revised manuscript is shown **in red**. All the significant changes in the manuscript are also highlighted in red.

Reviewer #1 (Remarks to the Author):

The manuscript "Local mapping of spatial correlations in the nanoscale diffusion dynamics of fluid biomembranes" by William J. Trewby, Mahdi Tavakol, and Kislun Voitchovsky present AFM-based local mapping of the fluidity of lipid bilayers at the nanoscale using an AFM-based approach.

The paper is interesting and well written, and I am in favor of publication.

Major concern: From what I read, I think the method is very powerful to characterize the nanoscale diffusion properties of lipids in a lipid membrane (or a biomembrane, but still only the lipids. I think simplicity the author think that too, as they say when they shear over a <7nm area/line, then they measure about 100 molecules (line 83). If they were to talk about membrane proteins that number would be 1. Thus, in very general terms, I dislike the use of the words biomembrane, as it implicates something much more complex with ~50% proteins packed and sugars, etc. Obviously, when the authors would land by coincidence (likely ~50% of the time in a real plasma membrane) on a 7nm (the typical size of a membrane protein) membrane protein, then they would probe the diffusion properties of that single molecule, which would be about an order of magnitude slower than the lipids. If it's a bigger protein, ever more so. Given that the method also relies on a 'contact mode' type of tip-sample interaction, a membrane protein would likely be plugged down on the mica surface at ~50pN constant force, and the diffusion would be essentially zero.

We thank the reviewer for these insightful comments. We agree that reliably measuring the diffusion of proteins would be challenging. The model developed in equation 1 allows for measuring larger objects than lipids: it only necessitates replacing the A_{lipid} by the area of the diffusing object. However, controlling the measurement on a plasma membrane packed with proteins and sugar would be extremely challenging, even with prior knowledge of the system. In such cases, the measurement can still take place, but its interpretation in terms of molecular mobility becomes questionable. Instead, the measurement needs to be interpreted in terms of the viscoelastic properties of the membrane-water interface. While less specific, it still carries important information about the interface's biophysical properties as

experienced by a molecule or a nanoscale object moving in close vicinity to the membrane.

In light of the reviewer's comment, the manuscript has been modified in multiple places.

First, the title has been changed to remove the claim of mapping the molecular diffusivity in biomembranes. Instead, we emphasise the highly localised viscoelasticity measurement. The quantification of the lipid mobility and possible lateral correlations require certain conditions to be met, as detailed later on in the paper.

In the introduction, the following sentence has been added to emphasise the viscoelastic nature of the measurement:

[...] In this mindset, some studies have attempted to quantify the viscoelastic properties of model lipid membranes³⁶, but reports of non-Newtonian behaviour remains controversial³⁷ with experimental limitations casting doubts on the results. To fully characterise the local viscoelastic forces and the resulting molecular dynamics of biological membranes, the appropriate spatial and temporal scales must then be interrogated simultaneously, with mechanical measurements offering a direct handle on forces. [...]

In the result section, the following text has been added to better clarify the fact the context of the measurements and lay the ground for the model as suitable only for viscous (fluid) membranes:

[...] The shear force can be intuitively understood as the resistance of the lipids trapped under the tip to the imposed shearing motion: if the effective diffusion of the group of lipids is slower or comparable to the imposed shear velocity, a net shear force arises. Consistently, the shear force increases with A_s and f_s at a given load (i.e. membrane indentation). Interpreting the associated phase is less straightforward. In rheology, a phase of 0° indicates a perfectly elastic (solid-like) behaviour while 90° indicates a purely viscous (fluid) behaviour. The phase is not defined if no shear force can be measured, here before the tip-membrane contact. Over the indentation region, the phase consistently increases towards a more viscous behaviour, but it always retains an elastic component. This behaviour is still observed when varying A_s and f_s . Upon rupture of the membrane (b), the tip eventually pins onto the substrate, reaching a phase of zero at higher loads (not shown). While this last point confirms the validity of the nano-rheology measurement, the apparent viscoelastic behaviour contradicts existing literature which suggests a purely viscous behaviour for simple fluid membrane^{36,37}. In contrast, interfacial water is notoriously sluggish due to correlated interactions⁵⁰⁻⁵³, offering a viscoelastic response to the shearing tip. Here, our observations can be rationalised by the AFM tip sensing the coupled dynamics of both, the lipids and the interfacial water in contact with the bilayer. As the tip indents the bilayer, the interfacial water between the tip and the membrane is progressively expelled, leading to a lipid-dominated, more viscous behaviour (Fig. 1b, c).

To test this interpretation, we conducted experiments aimed at varying the properties of the lipid membrane in a controlled manner without significantly affecting the interfacial water. We first probed the nano-rheological properties of a 1,2-Dimyristoyl-sn-glycero-3-phosphocholine (DMPC) bilayer just above its transition temperature, progressively increasing the temperature by steps of 4°C (Fig. 1d). The expected trend

emerges⁴⁸, with the shear force decreasing as the temperature increases for a given indentation, reflecting the enhanced lipid mobility. The phase is similar in all cases (within error) over the indentation region, qualitatively following the behaviour observed on DOPC. Alternatively, we added set amounts of cholesterol to the DOPC bilayer, thereby reducing its average lipid mobility while still ensuring a homogenous phase⁴⁹. This last experiment is more challenging, because it requires comparing different samples, measured in different sets of experiments. The results (Fig. 1e) still exhibit the expected trend with higher shear force measured for larger concentrations of cholesterol. The phase evolution also follows the trend, although with all curves overlapping within error. In all cases, it is straightforward to extract the shear and loss moduli of the sample, keeping in mind that fact that they contain information about both the lipid bilayer and the coupled interfacial water layer. Consistent with the proposed interpretation of the result, the loss (viscous) modulus captures better the expected membrane fluidity trends than the storage (elastic) modulus (see Supplementary section S4 for details).

Generally, all the experiments qualitatively support the proposed interpretation of the nano-rheology measurements whereby the shear force is dominated by the behaviour of the lipids at higher indentations while the phase carries mixed information about both the interfacial water and the lipids. To move beyond qualitative considerations, independent insight is needed. We therefore conducted computer simulations of the nano-rheological measurement process. [...]

In the new Discussion section, the following text has been added to emphasise the viscoelastic nature of the measurement:

We have developed a novel approach to quantify the in-plane molecular mobility of supported membranes with nanometre spatial resolution. Our technique uses nano-rheology to surpass the conventional temporal limitations of AFM to resolve in-plan dynamics and capture the local membrane dynamics. The approach offers some advantages over existing techniques, but it also has its limitations. The main advantages stem from the ‘mechanical’ nature of the measurement which provides a full viscoelasticity picture of the interfaces, probing both the lipids and the interfacial water. This measurement effectively mimics what a nanoscale object or molecule diffusing in the membrane’s vicinity experiences, probing the forces at play. It operates *in-situ*, locally, at the nanoscale and does not require any chemical or physical labelling of the sample. The ability to quantify the membrane’s viscoelastic response may be particularly useful for complex systems with significant local variations in mechanical properties and dynamics, something challenging with existing approaches. The technique’s limitations primarily come from the need for physical access to sample which must be immobilised. In biological systems, this limits measurement to outer membranes complexed if to be carried out on live cells, or to supported fragments of cells and organelles, as in this paper. Beyond the measurement itself, its interpretation of the results in terms of local membrane mobility hinges on assumptions that must be verified if to be valid. First, the sample needs to be fluid. Second, the Sackman-Evans model^{55,56} used in the model developed here is only valid for supported lipid membranes. A more general Saffman-Delbrück relationship exists for suspended membranes⁶⁴, but it is less favourable for achieving highly localised measurements due to logarithmic dependence on the area probed. Third, our model assumes

knowledge of the diffusing molecules species being probed and their respective size, as described in equation 1. While straightforward for lipid membranes, this becomes more challenging for plasma membranes which contain a high fraction of proteins and sugars. Finally, the derived diffusion coefficient values carry an error linked to the imperfect geometry of the system, compared to the idealised model: the tip apex is never perfectly spherical, and its measured size is to be taken as effective. However, the resulting error is systematic and affects each measurement with a given tip in the same way. The strength of the technique therefore lies in mapping relative variations across a sample, as illustrated in Figs. 3 and 4. On model biological membranes, the results quantify local variations in the lipid diffusivity even within apparently homogenous phases, revealing nanometre-scale correlations effects. Measurements on a native lens membrane also illustrate the potential of the technique to map functionally relevant nanoscale changes in molecular diffusivity across complex membranes. The ability of the technique to measure the viscoelastic response of the full system (membrane and interfacial water) shows that while the system is viscoelastic, this is primarily driven by the interfacial water with simple lipid membrane behaving mainly as viscous, hence enabling eq. 1 to be applied. This emphasises the interplay between the dynamics of the lipids and that of the water, as also clear from the interfacial storage (elastic) and loss (viscous) moduli (Supplementary section S4) and confirmed by the computer simulations.

In terms of spatiotemporal resolution, the technique is ideally placed to complement existing optical methods^{32–34} based on super-resolution and able to quantify the diffusion of labelled species in-vivo with typically ~1 ms and 50-100 nm resolution. For scanning probe based methods, we anticipate the technique to be particularly useful for quantifying the diffusion of lipids within fluid membranes, complementing existing approaches⁴¹ developed to track the conformational changes of proteins on a similar timescale.

In this context, I do not understand what the authors measure on the AQP0 patches in the eye lens membranes (cite Buzhysnky EMBO R 2007, and, Colom Nat Comm 2013, which both used AFM to image AQP0 patches in native eye lens membranes, the latter on cells). Indeed, as these authors could simply image the AQP0 patches and the authors here do that too, the diffusion of these large protein patches is essentially zero. Thus, what is the signal measured on these patches. I doubt that the presented shear method can provide a meaningful value for the diffusion of such systems.

The crux of the measurement presented in Fig. 4 is its demonstration of the significant variation in the diffusion coefficient over the fluid region of the membrane. In Fig. 4c, measurements locations 4 and 5 target mixed protein-lipid regions where no stable crystal assemblies are present with only the last two locations (6 and 7) probing the AQP0 crystal. We agree that in the case of these last two locations, the use of the mobility model is questionable. Replacing in equation 1 A_{lipid} for the area of a protein or a AQP0 tetramer would further reduce the derived diffusion coefficient by one or two orders of magnitude. Practically, this would not change much Fig. 4c where the associated diffusion coefficients are already near-zero when compared to that of the fluid regions. Indeed, as pointed out by the reviewer, in this region stable/immobile

protein crystals renders the diffusion negligible. Overall, this provides us with an example of breakdown in the applicability of equation 1 for the locations 6,7 which are intended as points of reference/comparison. We note that the tip can disrupt these stable crystals during the measurements, leading to subsequent fluid regions with a recovery in the apparent (assuming lipids) diffusion coefficient (see supplementary section 6, Fig. S14).

This was not clear in the manuscript and the text has now been clarified as follow:

[...] It is important to note that the diffusion coefficients in Fig. 4c were derived with eq. 1, implicitly assuming a fluid membrane and the lipids as the main diffusing specie probed. This is not necessarily true over protein-rich regions (positions 6, 7), where the area of a protein should be used instead of that of a lipid molecule in eq. 1. This substitution would further reduce the measured diffusion coefficient by an order of magnitude, consistent with the observed immobile protein assemblies. Over protein crystals, the assumption of a fluid membrane no longer applies and eq. 1 ceases to be valid. The tip can still be used to disrupt the protein assemblies, reversing the ordered domains in regions with higher molecular mobility, leading to the recovery of a finite diffusion coefficient (supplementary section S6). However, it is then no longer clear which molecular diffusion (proteins or lipids) is being probed. [...]

The suggested references have also been included in the text

Related to this, as the authors cite reference 38 (line 64), I feel that paper could receive a bit more consideration as the complementary method (interestingly with the same temporal $\sim 20\mu\text{s}$ resolution) to what is presented here. Indeed the topography based (tapping mode feedback operated method) by Heath Nat Comm 2018, is perfectly suited to characterize membrane protein dynamics. On the other hand, it cannot assess lipid dynamics, which the method presented by the authors clearly can.

We thank the reviewer for point out this complementarity which, we agree, supports the narrative of the paper. We have now added the following text where the reference is first introduced:

[...] Recent advances in the field have opened the milliseconds realm^{39,40}, with microsecond measurements possible in the direction perpendicular to the sample⁴¹ and dependent on a well-characterised topographical contrast. This enables detailed visualisation of the dynamics of proteins' motion, but the absence of lateral sensing precludes tracking the surrounding lipids dynamics. Here we address these limitations in the context of biomembranes viscoelastic measurements by combining AFM with a bespoke ultrafast in-plane actuator. [...]

We also added the following sentence near the end of the discussion section:

For scanning probe based methods, we anticipate the technique to be particularly useful for quantifying the diffusion of lipids within fluid membranes, complementing existing approaches⁴¹ developed to track the conformational changes of proteins on a similar timescale.

Thus, I recommend to rephrase the entire paper to make clear, as the methodology figure 1 shows, that the technique is mainly/only powerful to characterize the nanoscale diffusion dynamics of lipids, but that it has its limitations when it comes to larger objects such as membrane proteins.

We agree with the reviewer and have now implemented these recommendations, as detailed in our response to the previous points.

Minor concern: The method is based on a shear measurement, and the cantilevers have typically tremendously high torsional spring constants (for shear measurements). I find almost no or absence of information in the main text, and think some the cantilever (torsional) characterizations should be moved from supplementary information to main text. Also, discussing openly the torsional stiffness, and the detection limitations to the faint torsional deflections would be a gain to the manuscript, even if it points to some limitations.

The following sentence has been added to the main text of the paper in the first part of the results:

[...] To avoid stimulating flexural or torsional resonances of the measuring cantilever and minimise hydrodynamics effects, we use ultra-small cantilevers (USC-F1.5-k0.6, Nanoworld, Switzerland) with nominal planar dimensions of $(7 \times 3) \mu\text{m}^2$. Calculation of the cantilevers' torsional stiffness constant, k_ϕ , and sensitivity, γ_ϕ , were determined *in situ* from the torsional thermal spectrum in air and in solution⁴⁷ (see supplementary section S3 for details). The high torsional resonances make it easy to operate sub-resonance, but at the cost of high k_ϕ . The resulting sensitivity is typically in the range of a few piconewtons of lateral force. [...]

In summary I congratulate the authors to this development, and think it is most valuable. I do not request any additional experiments, but expect a thorough work-over of the text to focus and discuss what the technique is powerful for and where its limitations are.

We thank the reviewer for these encouraging comments and for pointing out the gaps in the presentation of our work. We believe this has helped us improve the paper.

Reviewer #2 (Remarks to the Author):

The manuscript reports an AFM-based study to measure the diffusion coefficient of lipids in supported lipid bilayer membranes. The manuscript presents a method based on the measurement of oscillatory shear forces. The experimental results are interpreted in terms of theory and coarse-grained MD simulations. The results illustrate the role of hydration water on the interfacial shearing behaviour. They illustrate also the potential of the technique to map local transitions in diffusivity in membranes.

This manuscript presents an excellent, timely and original method to measure the mobility of lipids at the nanoscale. It has the quality and relevance to be published in Nature Communications. Some revision is recommended.

Comments:

1 The AFM method would always measure shear forces. Therefore equation 1 would always provide a diffusion coefficient even on a solid surface without any diffusion species. The authors should clearly explain the conditions required to use properly eqn 1 for obtaining a diffusion coefficient.

We thank the reviewer for pointing this out. It was indeed a significant oversight on our part to omit this discussion. We have now extensively re-organised the paper and included two main discussions addressing this issue (in Results section and in the newly created Discussion section). The main point is that our nano-rheology measurements are general and provide information about the viscoelastic properties of the sample, regardless of its nature. However, in order to apply equation 1, several conditions have to be met: the membrane needs to be viscous (fluid), it needs to be supported and the size of the molecules probed by the tip must be known (here mainly lipids). The main elements of this discussion are reproduced hereafter to aid with the reviewing.

In results:

[...] The shear force can be intuitively understood as the resistance of the lipids trapped under the tip to the imposed shearing motion: if the effective diffusion of the group of lipids is slower or comparable to the imposed shear velocity, a net shear force arises. Consistently, the shear force increases with A_s and f_s at a given load (i.e. membrane indentation). Interpreting the associated phase is less straightforward. In rheology, a phase of 0° indicates a perfectly elastic (solid-like) behaviour while 90° indicates a purely viscous (fluid) behaviour. The phase is not defined if no shear force can be measured, here before the tip-membrane contact. Over the indentation region, the phase consistently increases towards a more viscous behaviour, but it always retains an elastic component. This behaviour is still observed when varying A_s and f_s . Upon rupture of the membrane (b), the tip eventually pins onto the substrate, reaching a phase of zero at higher loads (not shown). While this last point confirms the validity of the nano-rheology measurement, the apparent viscoelastic behaviour contradicts existing literature which suggests a purely viscous behaviour for simple fluid membrane^{36,37}. In contrast, interfacial water is notoriously sluggish due to correlated interactions^{50–53}, offering a viscoelastic response to the shearing tip. Here, our observations can be rationalised by the AFM tip sensing the coupled dynamics of both,

the lipids and the interfacial water in contact with the bilayer. As the tip indents the bilayer, the interfacial water between the tip and the membrane is progressively expelled, leading to a lipid-dominated, more viscous behaviour (Fig. 1b, c).

To test this interpretation, we conducted experiments aimed at varying the properties of the lipid membrane in a controlled manner without significantly affecting the interfacial water. We first probed the nano-rheological properties of a 1,2-Dimyristoyl-sn-glycero-3-phosphocholine (DMPC) bilayer just above its transition temperature, progressively increasing the temperature by steps of 4°C (Fig. 1d). The expected trend emerges⁴⁸, with the shear force decreasing as the temperature increases for a given indentation, reflecting the enhanced lipid mobility. The phase is similar in all cases (within error) over the indentation region, qualitatively following the behaviour observed on DOPC. Alternatively, we added set amounts of cholesterol to the DOPC bilayer, thereby reducing its average lipid mobility while still ensuring a homogenous phase⁴⁹. This last experiment is more challenging, because it requires comparing different samples, measured in different sets of experiments. The results (Fig. 1e) still exhibit the expected trend with higher shear force measured for larger concentrations of cholesterol. The phase evolution also follows the trend, although with all curves overlapping within error. In all cases, it is straightforward to extract the shear and loss moduli of the sample, keeping in mind that fact that they contain information about both the lipid bilayer and the coupled interfacial water layer. Consistent with the proposed interpretation of the result, the loss (viscous) modulus captures better the expected membrane fluidity trends than the storage (elastic) modulus (see Supplementary section S4 for details).

Generally, all the experiments qualitatively support the proposed interpretation of the nano-rheology measurements whereby the shear force is dominated by the behaviour of the lipids at higher indentations while the phase carries mixed information about both the interfacial water and the lipids. To move beyond qualitative considerations, independent insight is needed. We therefore conducted computer simulations of the nano-rheological measurement process. [...]

And in the discussion:

[...] Our technique uses nano-rheology to surpass the conventional temporal limitations of AFM to resolve in-plan dynamics and capture the local membrane dynamics. The approach offers some advantages over existing techniques, but it also has its limitations. The main advantages stem from the ‘mechanical’ nature of the measurement which provides a full viscoelasticity picture of the interfaces, probing both the lipids and the interfacial water. This measurement effectively mimics what a nanoscale object or molecule diffusing in the membrane’s vicinity experiences, probing the forces at play. It operates *in-situ*, locally, at the nanoscale and does not require any chemical or physical labelling of the sample. The ability to quantify the membrane’s viscoelastic response may be particularly useful for complex systems with significant local variations in mechanical properties and dynamics, something challenging with existing approaches. The technique’s limitations primarily come from the need for physical access to sample which must be immobilised. In biological systems, this limits measurement to outer membranes complexed if to be carried out on live cells, or to supported fragments of cells and organelles, as in this paper. Beyond the measurement itself, its interpretation of the results in terms of local

membrane mobility hinges on assumptions that must be verified if to be valid. First, the sample needs to be fluid. Second, the Sackman-Evans model^{55,56} used in the model developed here is only valid for supported lipid membranes. A more general Saffman-Delbrück relationship exists for suspended membranes⁶⁴, but it is less favourable for achieving highly localised measurements due to logarithmic dependence on the area probed. Third, our model assumes knowledge of the diffusing molecules species being probed and their respective size, as described in equation 1. While straightforward for lipid membranes, this becomes more challenging for plasma membranes which contain a high fraction of proteins and sugars. Finally, the derived diffusion coefficient values carry an error linked the imperfect geometry of the system, compared to the idealised model: the tip apex is never perfectly spherical, and its measured size is to be taken as effective. However, the resulting error is systematic and affects each measurement with a given tip in the same way. The strength of the technique therefore lies in mapping relative variations across a sample, as illustrated in Figs. 3 and 4. On model biological membranes, the results quantify local variations in the lipid diffusivity even within apparently homogenous phases, revealing nanometre-scale correlations effects. Measurements on a native lens membrane also illustrate the potential of the technique to map functionally relevant nanoscale changes in molecular diffusivity across complex membranes. The ability of the technique to measure the viscoelastic response of the full system (membrane and interfacial water) shows that while the system is viscoelastic, this is primarily driven by the interfacial water with simple lipid membrane behaving mainly as viscous, hence enabling eq. 1 to be applied. This emphasises the interplay between the dynamics of the lipids and that of the water, as also clear from the interfacial storage (elastic) and loss (viscous) moduli (Supplementary section S4) and confirmed by the computer simulations. [...]

2 AFM imaging was performed in amplitude modulation AFM. The authors might consider to include some general references to this popular AFM mode (for example, R. Garcia, R. Perez, Surf. Sci. 47, 197 (2002)).

This is an excellent suggestion. The reference has now been added to the paper, towards the end of the Introduction section.

[...] We acquire high-resolution images in amplitude modulation⁴² and subsequently quantify the local viscoelastic forces. [...]

Reviewer #3 (Remarks to the Author):

The article by Trewby et al. describe the construction of a new atomic force microscopy (AFM)-based method to measure diffusion coefficients in lipid membranes at high spatial resolution. After explaining the principle of the instrument and its characterization, the authors apply it to bilayers and a biological membrane. The article is well written, the work of good quality, and the conclusions are supported by the data. However, I think the article would fit better in a more specialized, possibly technical journal as I do not see the wider interest the way it is presented currently. The technique cannot be applied *in vivo*, and the biological membrane used is a reconstitution. The technique is not specific as it measures the diffusion coefficient of all membrane components simultaneously and cannot differentiate between different membrane embedded molecules. Therefore, while it allows measuring diffusion coefficients at high spatial and temporal resolution it cannot differentiate between different components. This is different from other techniques that offer the same spatiotemporal resolution but are also specific.

We thank the reviewer for this frank assessment of our work. We agree with their statement that the technique does not allow for molecular specificity and cannot be used *in-vivo* easily. It is also limited to samples where ‘physical’ access is possible and hence not intended to replace other techniques such as super-resolution approaches. Instead, its strength lies in its ability to probe the viscoelastic properties of the membrane-water interfaces, as would be experienced by a molecule or a nanoscale object moving in close vicinity to the membrane. If the membrane is viscous (fluid), then it is also possible to derive quantitative information about the lipids’ diffusivity. Measuring diffusivity is, in effect, a particular application of a more general viscoelastic measurement that stands even for static or gel-phase assemblies.

The ability to derive quantitative information about membrane’s viscoelasticity is highly relevant from a biophysical perspective and (to the best of our knowledge) not possible with other techniques. The measurement effectively captures the biomechanical signature of the membrane across multiple orders magnitude in terms of dynamics. Here the present results also help rationalise a lasting controversy in the field (see e.g. Harland, C. W., Miranda, B. J. & Parthasarathy, R. **Retraction of article “Phospholipid bilayers are viscoelastic”**. *Proc. Natl. Acad. Sci. U. S. A.* **108**, 14705–14705 (2011)) by pointing to the role played by interfacial water in the membrane dynamics.

We appreciate that none of this was clearly stated in the previous version of the manuscript and have therefore made significant changes throughout to better emphasise the viscoelastic nature of the measurement and the applicability (advantages and limitations) of the model we developed to extract the diffusion coefficient. This discussion is predominantly detailed in the in the Results section:

[...] The shear force can be intuitively understood as the resistance of the lipids trapped under the tip to the imposed shearing motion: if the effective diffusion of the group of lipids is slower or comparable to the imposed shear velocity, a net shear force arises. Consistently, the shear force increases with A_s and f_s at a given load (i.e. membrane indentation). Interpreting the associated phase is less straightforward. In

rheology, a phase of 0° indicates a perfectly elastic (solid-like) behaviour while 90° indicates a purely viscous (fluid) behaviour. The phase is not defined if no shear force can be measured, here before the tip-membrane contact. Over the indentation region, the phase consistently increases towards a more viscous behaviour, but it always retains an elastic component. This behaviour is still observed when varying A_s and f_s . Upon rupture of the membrane (b), the tip eventually pins onto the substrate, reaching a phase of zero at higher loads (not shown). While this last point confirms the validity of the nano-rheology measurement, the apparent viscoelastic behaviour contradicts existing literature which suggests a purely viscous behaviour for simple fluid membrane^{36,37}. In contrast, interfacial water is notoriously sluggish due to correlated interactions^{50–53}, offering a viscoelastic response to the shearing tip. Here, our observations can be rationalised by the AFM tip sensing the coupled dynamics of both, the lipids and the interfacial water in contact with the bilayer. As the tip indents the bilayer, the interfacial water between the tip and the membrane is progressively expelled, leading to a lipid-dominated, more viscous behaviour (Fig. 1b, c).

To test this interpretation, we conducted experiments aimed at varying the properties of the lipid membrane in a controlled manner without significantly affecting the interfacial water. We first probed the nano-rheological properties of a 1,2-Dimyristoyl-sn-glycero-3-phosphocholine (DMPC) bilayer just above its transition temperature, progressively increasing the temperature by steps of 4°C (Fig. 1d). The expected trend emerges⁴⁸, with the shear force decreasing as the temperature increases for a given indentation, reflecting the enhanced lipid mobility. The phase is similar in all cases (within error) over the indentation region, qualitatively following the behaviour observed on DOPC. Alternatively, we added set amounts of cholesterol to the DOPC bilayer, thereby reducing its average lipid mobility while still ensuring a homogenous phase⁴⁹. This last experiment is more challenging, because it requires comparing different samples, measured in different sets of experiments. The results (Fig. 1e) still exhibit the expected trend with higher shear force measured for larger concentrations of cholesterol. The phase evolution also follows the trend, although with all curves overlapping within error. In all cases, it is straightforward to extract the shear and loss moduli of the sample, keeping in mind that fact that they contain information about both the lipid bilayer and the coupled interfacial water layer. Consistent with the proposed interpretation of the result, the loss (viscous) modulus captures better the expected membrane fluidity trends than the storage (elastic) modulus (see Supplementary section S4 for details).

Generally, all the experiments qualitatively support the proposed interpretation of the nano-rheology measurements whereby the shear force is dominated by the behaviour of the lipids at higher indentations while the phase carries mixed information about both the interfacial water and the lipids. To move beyond qualitative considerations, independent insight is needed. We therefore conducted computer simulations of the nano-rheological measurement process.

And in the newly created Discussion section:

[...] Our technique uses nano-rheology to surpass the conventional temporal limitations of AFM to resolve in-plan dynamics and capture the local membrane dynamics. The approach offers some advantages over existing techniques, but it also has its limitations. The main advantages stem from the ‘mechanical’ nature of the

measurement which provides a full viscoelasticity picture of the interfaces, probing both the lipids and the interfacial water. This measurement effectively mimics what a nanoscale object or molecule diffusing in the membrane's vicinity experiences, probing the forces at play. It operates *in-situ*, locally, at the nanoscale and does not require any chemical or physical labelling of the sample. The ability to quantify the membrane's viscoelastic response may be particularly useful for complex systems with significant local variations in mechanical properties and dynamics, something challenging with existing approaches. The technique's limitations primarily come from the need for physical access to sample which must be immobilised. In biological systems, this limits measurement to outer membranes complexed if to be carried out on live cells, or to supported fragments of cells and organelles, as in this paper. Beyond the measurement itself, its interpretation of the results in terms of local membrane mobility hinges on assumptions that must be verified if to be valid. First, the sample needs to be fluid. Second, the Sackman-Evans model^{55,56} used in the model developed here is only valid for supported lipid membranes. A more general Saffman-Delbrück relationship exists for suspended membranes⁶⁴, but it is less favourable for achieving highly localised measurements due to logarithmic dependence on the area probed. Third, our model assumes knowledge of the diffusing molecules species being probed and their respective size, as described in equation 1. While straightforward for lipid membranes, this becomes more challenging for plasma membranes which contain a high fraction of proteins and sugars. Finally, the derived diffusion coefficient values carry an error linked the imperfect geometry of the system, compared to the idealised model: the tip apex is never perfectly spherical, and its measured size is to be taken as effective. However, the resulting error is systematic and affects each measurement with a given tip in the same way. The strength of the technique therefore lies in mapping relative variations across a sample, as illustrated in Figs. 3 and 4. On model biological membranes, the results quantify local variations in the lipid diffusivity even within apparently homogenous phases, revealing nanometre-scale correlations effects. Measurements on a native lens membrane also illustrate the potential of the technique to map functionally relevant nanoscale changes in molecular diffusivity across complex membranes. The ability of the technique to measure the viscoelastic response of the full system (membrane and interfacial water) shows that while the system is viscoelastic, this is primarily driven by the interfacial water with simple lipid membrane behaving mainly as viscous, hence enabling eq. 1 to be applied. This emphasises the interplay between the dynamics of the lipids and that of the water, as also clear from the interfacial storage (elastic) and loss (viscous) moduli (Supplementary section S4) and confirmed by the computer simulations. In terms of spatiotemporal resolution, the technique is ideally placed to complement existing optical methods³²⁻³⁴ based on super-resolution and able to quantify the diffusion of labelled species *in-vivo* with typically ~1 ms and 50-100 nm resolution. For scanning probe based methods, we anticipate the technique to be particularly useful for quantifying the diffusion of lipids within fluid membranes, complementing existing approaches⁴¹ developed to track the conformational changes of proteins on a similar timescale.

Detailed comments

Line 58: There are articles which have shown to be able to measure at high spatiotemporal scales (see Eggeling et al. Nature 2009, Xiang et al. Nature Methods 2020; Sankaran et al. Nature Communications 2021). These methods could be compared to the current method in terms of spatiotemporal resolution, specificity, and usable sample types. Especially single particle tracking could be discussed as that allows measuring of a single molecule at similar spatial resolution.

This is an interesting idea. Eggeling et al. use far-field detection STED in conjunction with fluorescently labelled lipids. The spatial resolution of STED is ~50 nm and using fluorescence autocorrelation over this area allows quantification of the associated 'transit time' for single labelled molecules. The spatial resolution is achieved by dilute markers concentrations, as common for single molecule optical techniques. The typical timescales identified are in the order of 10 ms, but autocorrelation is possible down to sub-0.1 ms. Xiang et al. develop a single molecule technique to track the diffusion of labelled molecules across whole cells with ~1 ms, ~100 nm resolution. The power of the technique comes from its ability to map the diffusion for whole cells *in-vivo*. Finally, Sankaran et al., used GPU-based analysis to exploit the best possible temporal and spatial resolution of the optical camera used for super-resolution measurements. This allows them, through use of a multiparametric resolution approach, to achieve ~2 ms, ~100 nm spatiotemporal resolution, also mapping organisms *in-vivo*.

In comparison, our technique does not rely on fluorescent labels and can derive comprehensive viscoelastic information (diffusion being one subset of this information on fluid membranes) and with higher spatiotemporal resolution (~10 nm, ~20 μ s). However, measurements being local (near field), they have to be conducted sequentially if to map a membrane, and they require physical access. This makes *in-vivo* measurements far more challenging. As such, the technique is not meant as an alternative to optical techniques, but rather a complement.

Following this comment by the reviewer, we have now added elements of the comparison as well as our vision for the technique in the discussion section:

In terms of spatiotemporal resolution, the technique is ideally placed to complement existing optical methods^{32–34} based on super-resolution and able to quantify the diffusion of labelled species *in-vivo* with typically ~1 ms and 50-100 nm resolution. For scanning probe based methods, we anticipate the technique to be particularly useful for quantifying the diffusion of lipids within fluid membranes, complementing existing approaches⁴¹ developed to track the conformational changes of proteins on a similar timescale.

What is the actual tip radius used (2nm similar to the simulated spheres?) and how much do the results depend on this factor, i.e. do the authors experimentally recover the same mean diffusion coefficient and standard deviation when the tip radius varies?

This is an important point. Commercial tip manufacturers provide indicative radii (typically 2 nm to 20 nm), but the actual size can vary between tips due to the manufacturing process. We therefore characterised the tips by electron microscopy

ex-situ to determine the radius to be used in equation 1. We tested the impact of changes in the tip size and found that the measurements are remarkably robust against size variations as mentioned in the text:

[...] However, the compensating relationship between tip radius and indentation depth for any given load means the data are surprisingly robust to changes in R_{tip} [...]

The physical principles behind this robustness, backed by experimental evidence with tips exhibiting different sizes, are explained in detail in the supplementary section S5 including Fig. S13.

Still, the tip is never perfectly spherical as assumed in the model, and determining an effective the tip radius carries some uncertainty. As a result, the measured value of the diffusion coefficient often carries a systematic error. This error does not affect relative variations in the measured diffusion coefficient across a given sample. As such, the technique is best suited to identifying fine variations across a sample rather than for comparison of different sets of experiments (different samples or tips). This is now discussed in the paper's Discussion section as follows:

[...] Finally, the derived diffusion coefficient values carry an error linked the imperfect geometry of the system, compared to the idealised model: the tip apex is never perfectly spherical, and its measured size is to be taken as effective. However, the resulting error is systematic and affects each measurement with a given tip in the same way. The strength of the technique therefore lies in mapping relative variations across a sample, as illustrated in Figs. 3 and 4. [...]

A very interesting result is that the authors see a bimodal distribution of the diffusion coefficient in Lo domains. Is there an explanation for that and does this mean the Lo domains are not homogeneous but are rather a mixture of Lo and Ld domains? The authors might also want to compare their diffusion coefficients in Lo domains to what has been measured before as the results seem to be quite high (see next point).

Prompted by the comment of the reviewer, we carefully re-analysed the data. Given the sequential nature of the measurement, our concern was that the bimodal distribution could be induced by some positional drift in the measurements close to the Lo/Ld boundary resulting in some curves being incorrectly ascribed to Lo or Ld. To be as objective as possible, we compared the standard AFM images acquired immediately before and after the viscoelasticity mapping, and only retained as Lo (respectively Ld) data the curves acquired over domains that overlapped between the two images. We excluded any contentious measurement where the overlap was not clear. Unsurprisingly, this strategy identified primarily border regions as ambiguous (see supplementary section 3.3.3 and Fig. S7 for details). The re-calculated distribution exhibits the same wider diffusion distribution over the Lo domains, but no bimodal behaviour. This suggests that the bimodal peak was an error resulting from incorrectly including Ld curves in the statistics.

Separately, we find the breadth of the Lo distribution (compared to the Ld distribution) interesting because it suggests a less homogenous phase at the nanoscale and lateral

correlations, despite a consistent average picture. This is further explored in our answer to the reviewer's later comments.

We corrected Fig. 3 with the detailed analysis now in SI (supplementary section 3.3.3). We also added the following sentence to the main text:

[...] This spread represents genuine nanoscale variations of the diffusion across a given domain and not an experimental error. The spread is exacerbated over the L_o domains due to the lower molecular mobility rendering the measurement more sensitive to local molecular variations. This dependence on local variations suggests some possible spatial correlations in the diffusivity across the membrane. [...]

We answer the point about the slightly high diffusion coefficients with the next question.

Line 151 and Fig. 1: The diffusion coefficient for DOPC is on the high side. It exceeds six of the cited DOPC results significantly and is only smaller than one exceptional measurement that determines a diffusion coefficient of 8 $\mu\text{m}^2/\text{s}$, i.e. at least a factor of two higher compared to all other measurements. So it is difficult to argue that the authors' results lie within the bulk of the measurements. And measurements on membranes of ternary mixtures but also on biological membranes are quite high, although they show more variability. This raises the question of how well diffusion coefficients can be measured and whether there is a bias. It would be interesting to see measurements on multiple single component bilayers and perhaps at different temperature, gel and liquid phases, to demonstrate the diffusion coefficient is unbiased and show the resolution the technique can achieve. These measurements could also be used to demonstrate that the local correlations in diffusion coefficients are reliable and follow the expected trends in liquid and gel phases.

As noted by the reviewer, the diffusion coefficient values are relatively high across the board, for both L_o and L_d domains. We believe this systematic error points to the uncertainty with determining precisely the size and shape of the shearing tip. As discussed in previous answers, the model assumes a perfectly spherical tip which we know not to be the case usually. This can induce a systematic error that could be corrected by a simple rescaling, if calibrating the measurement over a known sample. However, the emphasis is rather placed on relative variations across a given sample, as mapped in Fig. 3a. Such variations remain valid regardless of any systematic bias, as now discussed in the Discussion section.

That being said, we find that there is a significant variability in the reported diffusion coefficients in the literature, as illustrated by Table S1 in supplementary information. Some of the measurements in the references proposed by the reviewer as point of comparison for our technique reported values as high as 30 $\mu\text{m}^2/\text{s}$, although not with supported membranes.

The reviewer makes an excellent point about finding some test system where the lipid mobility can be varied systematically and controllably. We heeded their advice and conducted two different tests. First, we conducted measurements over DMPC at different temperature close to –but above– its transition temperature when supported. The variation in diffusion coefficient with temperature close the transition is well

documented (see e.g. Scomparin, C., Lecuyer, S., Ferreira, M., Charitat, T. & Tinland, B. Diffusion in supported lipid bilayers: Influence of substrate and preparation technique on the internal dynamics. *Eur. Phys. J. E* **28**, 211–220 (2009)). The results (Fig. 1d) show that the technique can clearly distinguish between variations in temperature as small as 4 °C. Optical measurements on the same system suggest variations in the diffusion coefficient in the order of 10% over such temperature changes, but with an experimental uncertainty preventing resolution of such small temperature steps as probed. The DMPC measurements represent the ideal test for the technique because it can be conducted within a same set of measurements by simply varying the temperature.

Second, we tackled a more challenging system: DOPC with increasing concentrations of cholesterol. While it is well-known that cholesterol decreases the average diffusivity of the lipids due to the formation of hydrogen bonds with the PC headgroups, the nanoscale picture is still not clear with even FRAP measurements on SLBs reported only recently (Zhang, Y., Li, Q., Dong, M. & Han, X. Effect of cholesterol on the fluidity of supported lipid bilayers. *Colloids and Surfaces B: Biointerfaces* **196**, 111353 (2020)). Here the measurements are particularly challenging because it requires comparing different samples and hence different sets of experiments. We retained the same tip throughout to maximise comparability. Again, the results show the expected evolution, with larger shear forces observed for higher cholesterol content at a given load. Interestingly, the phase also exhibits differences between samples, mainly for DMPC and to some extent for the DOPC+Chol experiments, confirming that the viscoelasticity of the membrane-water system also changes. The different measurements on test synthetic membranes (DOPC vs frequency, DMPC vs temperature, DOPC+chol vs chol concentration) also illustrate the interplay between the behaviour of the interfacial water and that of the bilayer: the water dominates the elastic response of the system whereas the membrane is viscous; a key requirement to applying our model for quantifying diffusivity. This is visible when examining the storage and loss moduli of the bilayers (supplementary section S4, Fig. S11) where the expected membrane evolution is always clearer in the loss (viscous) modulus.

The new results have been incorporated in the revised manuscript, changing the flow of the narrative and focusing first on the interpretation of the viscoelastic measurements before moving on the developing the model for interpreting the data over viscous (fluid) membranes in terms of diffusivity.

Fig. 1 has been updated:

Fig. 1 | Nanoscale tracking of molecular mobility with AFM nano-rheology. Schematic of apparatus (a) (not to scale), comprising a high-frequency shear piezo actuator, mounted directly onto the AFM scanner and able to apply a controllable shear amplitude A_s . An external lock-in amplifier measures the shear (in-plane) force, F_s , and phase, θ_s , as experienced by the nanoscale tip for a given shear frequency, f_s . The entire cantilever, sample and substrate are immersed in the imaging fluid. The distance between the tip and the membrane as well as the normal force F_N applied by the tip onto the sheared membrane are carefully controlled. In a spectroscopy measurement on a supported DOPC lipid membrane (b), the tip approaches the membrane from the bulk solution. Away from the membrane, no normal force is applied, and no shear force measured aside from hydrodynamic drag. We hence set $F_N = F_s = 0$ as a reference and the shear phase is undefined (semi-transparent region). As the tip begins to interact and press on the membrane (yellow highlight) F_N increases, but the magnitude of F_s depends on the shear velocity. At 1 kHz, $F_s \sim 0$ and θ_s is still undefined (red), but F_s and $\theta_s > 0$ at 50 kHz (black), with θ_s increasing with F_N . Once F_N surpasses a particular threshold, the membrane ruptures and the tip eventually becomes pinned to the substrate underneath. (c) In the membrane indentation region (yellow highlight in (b)), the shear force increases with both the shear amplitude (for fixed f_s) and frequency (for fixed amplitude, A_s), indicating a consistent increase of F_s with the shearing velocity v_s . Similarly to (b), θ_s tends to increase with F_N in any given situation, indicating a transition from an elastic-dominated response to a more viscous-dominated behaviour of the bilayer-water system. (d) The sensitivity of the measurement can be tested on a pure DMPC bilayer at different temperatures above its gel-fluid transition. Qualitatively, the behaviour is as for DOPC, but the F_s decreases with increasing temperature, indicating a more fluid membrane⁴⁸. Similarly (e), adding cholesterol to a DOPC membrane reduces its molecular mobility⁴⁹, increasing the measured shear force. $A_s = 5$ nm in (b) and 7 nm in (c, lower panel), $f_s = 30$ kHz in (c, upper panel). In (c-e), the curves and shading represent the mean of $n > 15$ force curves and one standard deviation respectively.

The text of the Results section has also been adapted accordingly:

The shear force can be intuitively understood as the resistance of the lipids trapped under the tip to the imposed shearing motion: if the effective diffusion of the group of lipids is slower or comparable to the imposed shear velocity, a net shear force arises. Consistently, the shear force increases with A_s and f_s at a given load (i.e. membrane indentation). Interpreting the associated phase is less straightforward. In rheology, a phase of 0° indicates a perfectly elastic (solid-like) behaviour while 90° indicates a purely viscous (fluid) behaviour. The phase is not defined if no shear force can be measured, here before the tip-membrane contact. Over the indentation region, the phase consistently increases towards a more viscous behaviour, but it always retains an elastic component. This behaviour is still observed when varying A_s and f_s . Upon rupture of the membrane (b), the tip eventually pins onto the substrate, reaching a phase of zero at higher loads (not shown). While this last point confirms the validity of the nano-rheology measurement, the apparent viscoelastic behaviour contradicts existing literature which suggests a purely viscous behaviour for simple fluid membrane^{36,37}. In contrast, interfacial water is notoriously sluggish due to correlated interactions⁵⁰⁻⁵³, offering a viscoelastic response to the shearing tip. Here, our observations can be rationalised by the AFM tip sensing the coupled dynamics of both, the lipids and the interfacial water in contact with the bilayer. As the tip indents the bilayer, the interfacial water between the tip and the membrane is progressively expelled, leading to a lipid-dominated, more viscous behaviour (Fig. 1b, c).

To test this interpretation, we conducted experiments aimed at varying the properties of the lipid membrane in a controlled manner without significantly affecting the interfacial water. We first probed the nano-rheological properties of a 1,2-Dimyristoyl-sn-glycero-3-phosphocholine (DMPC) bilayer just above its transition temperature, progressively increasing the temperature by steps of 4°C (Fig. 1d). The expected trend emerges⁴⁸, with the shear force decreasing as the temperature increases for a given indentation, reflecting the enhanced lipid mobility. The phase is similar in all cases (within error) over the indentation region, qualitatively following the behaviour observed on DOPC. Alternatively, we added set amounts of cholesterol to the DOPC bilayer, thereby reducing its average lipid mobility while still ensuring a homogenous phase⁴⁹. This last experiment is more challenging, because it requires comparing different samples, measured in different sets of experiments. The results (Fig. 1e) still exhibit the expected trend with higher shear force measured for larger concentrations of cholesterol. The phase evolution also follows the trend, although with all curves overlapping within error. In all cases, it is straightforward to extract the shear and loss moduli of the sample, keeping in mind that fact that they contain information about both the lipid bilayer and the coupled interfacial water layer. Consistent with the proposed interpretation of the result, the loss (viscous) modulus captures better the expected membrane fluidity trends than the storage (elastic) modulus (see Supplementary section S4 for details).

Generally, all the experiments qualitatively support the proposed interpretation of the nano-rheology measurements whereby the shear force is dominated by the behaviour of the lipids at higher indentations while the phase carries mixed information about both the interfacial water and the lipids. To move beyond qualitative considerations, independent insight is needed. We therefore conducted computer simulations of the nano-rheological measurement process.

Fig. 3d: The central data points seem to fall on an ellipse that is not centered at the origin and that has a much stronger inclination than the fitted 0.158 gradient indicated. I assume the expected correlation cloud should be symmetrical to the origin in respect to the diffusion coefficient? And what does the stronger inclination imply? That diffusion coefficients are strongly correlated? And what are the differences in these correlations within the Lo and the Ld phase? Here, too, it would be very interesting to see data on more samples, especially more ordered or gel phases, to show that the correlation length reflect bilayer properties.

A larger gradient would indeed indicate a stronger lateral correlation of the diffusion coefficients within the dataset considered. By eye the gradient indeed appears steeper than indicated by the fit, but having carefully re-checked our analysis we can confirm that the reported gradient is indeed correct (0.158). The appearance of a steeper gradient or off-centre for the ellipse is likely due to visual effects in the interpretation of the datapoints: if only the centre-most datapoints are considered (between -1 and +1), then the gradient would be closer to 0.3 suggesting a stronger correlation. Data points further from the centre and in the lower right quadrant likely skew the gradient to a lower value, but these points must be considered for unbiased statistics. Additionally, the previous black and white representation of the datapoints may have hindered visualisation of the local variations in density. To address this last point, we have now converted the black and white representation to a heatmap in the revised Fig. 3f. To us, the ellipse seems well-centred albeit with a gradient still appearing lower than a visual fit would place it. The ellipse is not symmetrical, but this is not expected since correlations between locations of lower mobilities do not necessarily mirror those over regions with higher mobilities.

Generally, our analysis aims to distinguishing two things: the strength of the correlation (i.e. the value of the gradient) and the correlation's statistical significance. The statistical significance is important given the broad distribution of datapoints in Fig. 3f: in principle a steep gradient could be down to a fluke while a small gradient may be statistically highly significant. We therefore randomised our dataset and repeated the correlation test, confirming a high statistical significance ($p < 0.01$) despite the apparent modest gradient.

Finally, the reviewer suggested a finer analysis of the correlation, distinguishing results from Lo and Ld domains. Given the reduced diffusivity over the Lo domains, we would expect a higher degree of spatial correlation compared to Ld domains. This is also supported by the broader histogram distribution in Fig. 3d. Having conducted the analysis over the two domains, we indeed find a correlation ~ 3 times higher on Lo regions compared to Ld regions. Both regions return a positive spatial correlation that is statistically significant although the gradient values are still low with larger variability over the Ld domains ($I_{Lo} = 0.146 \pm 0.007$, $I_{Ld} = 0.051 \pm 0.049$). The result is consistent and encouraging, but we feel that it should be taken with some caution given the relatively small number of relevant datapoints in each case. This is exacerbated by the fragmented geometry of the domains which further limits the number of locations where all nearest neighbours are available. In fact, we suspect the boundary regions to be the most spatially correlated in terms of diffusion, but no meaningful statistics could be derived from the available data.

The following text has now been added to the paper, where discussing the correlations in Fig. 3:

Here we find a positive correlation, with $I = 0.158 \pm 0.042$ ($n = 816$), as illustrated graphically in Fig. 2f, and with a strong statistical significance ($p < 0.0025$). This confirms that the local diffusion is globally dependent on the immediate nanoscale environment (see supplementary section S1.3 for more details on the test). Conducting the same analysis separately over the L_O and L_D domains confirm that spatial correlations in the diffusion exist over both domains, but with degree of correlation three times higher for L_O regions ($I_{LO} = 0.146 \pm 0.007$, $n_{LO} = 291$; $I_{LD} = 0.051 \pm 0.049$, $n_{LD} = 461$). The results are statistically significant in both cases, but the fragmented geometry of the domains (see supplementary section S3.3.3) and the reduced data subset available for each region renders the analysis less robust than over the entire image. The boundary regions may also play an important role in setting lateral correlations, although not easily addressable here due to the small size of the relevant region. Overall, the results confirm that the spread in diffusion values over the different domains (Fig. 3d) reflects some spatial correlations within the membrane.